Arktocara yakataga, a new fossil odontocete (Mammalia, Cetacea) from the Oligocene of Alaska and the antiquity of Platanistoidea

Boersma Alexandra T. boersma.alex@gmail.com 1 2
Pyenson Nicholas D. 1 3
1 Department of Paleobiology, National Museum of Natural History, Smithsonian Institution , Washington , D.C. , United States of America
2 College of Extended Education, California State University, Monterey Bay , Seaside , CA , United States of America
3 Departments of Mammology and Paleontology, Burke Museum of Natural History and Culture , Seattle , WA , United States of America
Thewissen J
Electronic publication date: 2016 Aug 16
Publication date: 2016
Volume: 4
Electronic Location ID: e2321
Received 2016 May 16; Accepted 2016 Jul 13
Copyright: ©2016 Boersma and Pyenson
Copyright year: 2016
Copyright holder: Boersma and Pyenson
License: This is an open access article distributed under the terms of the Creative Commons Attribution License, which permits unrestricted use, distribution, reproduction and adaptation in any medium and for any purpose provided that it is properly attributed. For attribution, the original author(s), title, publication source (PeerJ) and either DOI or URL of the article must be cited.
License URL: https://creativecommons.org/licenses/by/4.0/

Keywords: Cetacea, River dolphins, Fossil record, Neogene, Evolution, Platanistoidea, Oligocene, Pan-Platanista

Funding: The Smithsonian Institution The research of both ATB and NDP was funded by the Smithsonian Institution, its Remington Kellogg Fund, and with support from the Basis Foundation. The funders had no role in study design, data collection and analysis, decision to publish, or preparation of the manuscript.

==============================
The diversification of crown cetacean lineages (i.e., crown Odontoceti and crown Mysticeti) occurred throughout the Oligocene, but it remains an ongoing challenge to resolve the phylogenetic pattern of their origins, especially with respect to stem lineages. One extant monotypic lineage, Platanista gangetica (the Ganges and Indus river dolphin), is the sole surviving member of the broader group Platanistoidea, with many fossil relatives that range from Oligocene to Miocene in age. Curiously, the highly threatened Platanista is restricted today to freshwater river systems of South Asia, yet nearly all fossil platanistoids are known globally from marine rocks, suggesting a marine ancestry for this group. In recent years, studies on the phylogenetic relationships in Platanistoidea have reached a general consensus about the membership of different sub-clades and putative extinct groups, although the position of some platanistoid groups (e.g., Waipatiidae) has been contested. Here we describe a new genus and species of fossil platanistoid, Arktocara yakataga, gen. et sp. nov. from the Oligocene of Alaska, USA. The type and only known specimen was collected from the marine Poul Creek Formation, a unit known to include Oligocene strata, exposed in the Yakutat City and Borough of Southeast Alaska. In our phylogenetic analysis of stem and node-based Platanistoidea, Arktocara falls within the node-based sub-clade Allodelphinidae as the sister taxon to Allodelphis pratti. With a geochronologic age between ∼29–24 million years old, Arktocara is among the oldest crown Odontoceti, reinforcing the long-standing view that the diversification for crown lineages must have occurred no later than the early Oligocene.

Introduction

Multiple lines of evidence point to the Oligocene epoch as an important period for the origin and early evolutionary history of crown group Cetacea (Fordyce, 2003). This interval, from about ∼34 to ∼23 million years ago, represents the origin of all extant lineages of cetaceans, including crown members of Mysticeti and Odontoceti, as inferred from molecular clock divergence estimates (McGowen, Spaulding & Gatesy, 2009), and fossil data (Geisler et al., 2011; Marx & Fordyce, 2015). Fossil cetaceans from this time are relatively less well known than Neogene ones for several reasons, including a lack of available rock outcrops, and insufficient study and publication on material housed in museum and personal collections (Uhen & Pyenson, 2007). The description of new cetacean taxa from the Oligocene can therefore be significant in resolving phylogenetic patterns of divergences among crown and stem groups, especially within Odontoceti.

Oligocene fossil cetaceans have played an important role in understanding the evolutionary history of Platanistoidea, once a large group of cosmopolitan marine odontocetes, but now represented by only one freshwater river species: Platanista gangetica (Lebeck, 1801), found in the Indus, Ganges-Brahmaputra-Megna and Karnaphuli-Sangu river systems of Southeast Asia. The concept of Platanistoidea has changed drastically since it was first defined by Simpson (1945) to include only one nominal family, Platanistidae, consisting of the four extant river dolphin lineages (Platanista Wagler, 1830, Inia D’Orbigny, 1834, Lipotes Miller, 1918, and Pontoporia Gray, 1846) and their closest fossil relatives. Muizon (Muizon, 1984; Muizon, 1987; Muizon, 1988a) later suggested a polyphyletic interpretation of the river dolphin lineages, modifying Simpson’s (1945) concept of Platanistoidea to include Platanista as the only crown taxon, with Inia, Lipotes and Pontoporia more closely related to Delphinoidea. This concept is consistent with the results from more recent phylogenetic analyses, especially those using molecular datasets (see Geisler et al., 2011 for a review).

Currently, phylogenetic studies of Platanistoidea have reached a general consensus about the inclusion three groups (two of which are completely extinct): Allodelphinidae, Squalodelphinidae and Platanistidae (including extant Platanista). The inclusion of Squalodontidae and Waipatiidae in Platanistoidea, as suggested by Muizon (1984), Muizon (1987), Muizon (1988a), Muizon (1994) and Fordyce (1994), have been more heavily debated (Lambert, Bianucci & Urbina, 2014; Tanaka & Fordyce, 2015a). Until now, no comprehensive phylogenetic analysis has used a taxonomic sampling that includes all putative platanistoid lineages, along with appropriate outgroups, such as Delphinida and stem Odontoceti (Tanaka & Fordyce, 2015a; Kimura & Barnes, 2016; Lambert, Bianucci & Urbina, 2014; Geisler & Sanders, 2003; Geisler et al., 2011).

Here we describe Arktocara yakataga, a new genus and species of Allodelphinidae, collected in 1951 from the Poul Creek Formation in the Yakutat City and Borough of Southeast Alaska (Figs. 1–6). Arktocara yakataga is the most northern platanistoid yet reported, and with an estimated mid-Oligocene age (possibly Rupelian to Chattian), it is both the oldest allodelphinid and among the oldest crown Odontoceti known. The results of our phylogenetic analysis continue to support a monophyletic Platanistoidea, along with traditional sub-clades (including a monophyletic Allodelphinidae), which underscores the importance of Oligocene cetaceans in documenting the early diversification of crown Cetacea.

Figure 1 Map of type locality for Arktocara yakataga (USNM 214830).

(A) a map of the state of Alaska, showing major Alaskan cities. (B) simplified geologic map of the Yakutat City and Borough based on the USGS 1971 map by Don J. Miller (available at http://ngmdb.usgs.gov/Prodesc/proddesc_9402.htm). All exposures of the Poul Creek Formation in the Yakutat City and Borough (orange), are potential type localities for Arktocara yakataga (USNM 214830). Yellow represents exposures from all other lithological units, not mapped here.

Figure 2 Skull of Arktocara yakataga (USNM 214830) in dorsal view.

(A) Illustrated skull with low opacity mask, interpretive line art, and labels for skull elements. Dotted lines indicate uncertainty of sutures and hatched lines indicate sediment obscuring the fossil. (B) photograph of skull in dorsal view, photography by James Di Loreto, Smithsonian Institution.

Figure 3 Skull of Arktocara yakataga (USNM 214830) in ventral view.

(A) Illustrated skull with low opacity mask, interpretive line art, and labels for skull elements. Dotted lines indicate uncertainty of sutures, dashed lines indicate broad morphological features, and hatched lines indicate sediment obscuring the fossil. (B) photograph of skull in ventral view, photography by James Di Loreto, Smithsonian Institution.

Figure 4 Skull of Arktocara yakataga (USNM 214830) in left (A, B) and right (C, D) lateral views.

(A) Illustrated left lateral view of skull with low opacity mask, interpretive line art, and labels for skull elements. Dotted lines indicate uncertainty of sutures and hatched lines indicate sediment obscuring the fossil. (B) photograph of skull in left lateral view, photography by James Di Loreto, Smithsonian Institution. (C) Illustrated right lateral view of skull with low opacity mask, interpretive line art, and labels for skull elements. (D) photograph of skull in right lateral view, photography by James Di Loreto, Smithsonian Institution.

Figure 5 Skull of Arktocara yakataga (USNM 214830) in anterior (A, B) and posterior (C, D) views.

(A) Illustrated skull in anterior view with low opacity mask, interpretive line art, and labels for skull elements. Dotted lines indicate uncertainty of sutures and hatched lines indicate sediment obscuring the fossil. (B) photograph of skull in anterior view, photography by James Di Loreto, Smithsonian Institution. (C) Illustrated skull in posterior with low opacity mask, interpretive line art, and labels for skull elements. (D) photograph of skull in posterior view, photography by James Di Loreto, Smithsonian Institution.

Figure 6 Skull details & sinus system of Arktocara yakataga (USNM 214830).

(A) Illustrated detail of right ventrolateral skull with low opacity mask, interpretive line art, and labels for skull elements. Dotted lines indicate uncertainty of sutures and hatched lines indicate sediment obscuring the fossil. Arrows indicate anatomical direction, with a, anterior and l, left lateral. (B) Illustrated detail of right ventrolateral skull with low opacity mask, interpretive line art, and blue highlighted areas indicating fossa for lobes of the sinus system.

Materials and Methods

Digital methods

The holotype of Arktocara yakataga was scanned using Nikon Metrology’s combined 225/450 kV microfocus X-ray and computed tomography (CT) walk-in vault system at Chesapeake Testing in Belcamp, Maryland, USA. Using this vault CT scanner system, we collected CT slices at 0.63 mm, resulting in three-dimensional reconstruction increments of 0.30 mm. We mounted the holotype skull vertically in the vault CT scanner system, with the posterior side down to minimize scanning width. Also, we collected CT scan data for the right periotic (YPM 13408) of Allodelphis pratti Wilson, 1935 using their Nikon Metrology’s 225 kV microfocus X-ray CT cabinet system. The DICOM files that this produced were processed in Mimics (Materialise NV, Leuven, Belgium) to create a 3D model of the Arktocara cranium that will be available for viewing and download on the Smithsonian X 3D website (http://3d.si.edu). These 3D files, along with the original DICOM files, are also archived at Zenodo (http://zenodo.org) at the following DOI: 10.5281/zenodo.51363. While the CT data were useful for making 3D models of the holotype, the density of the fossil material prevented the CT scanner from clearly determining internal morphology of the skull, failing to clarify indistinct sutures between skull elements.

Phylogenetic analysis

We tested the phylogenetic placement of Arktocara yakataga using Tanaka & Fordyce (2015a)’s Odontoceti matrix, adapted from Murakami et al.’s (2012) original version. Tanaka & Fordyce (2015a)’s version of this matrix consisted of 292 morphological characters and 83 operational taxonomic units (OTUs), including the fossil platanistoids Notocetus vanbenedeni Moreno, 1892, Phocageneus venustus Leidy, 1869, Squalodon calvertensis Kellogg, 1923, Waipatia maerewhenua Fordyce, 1994, Zarhachis flagellator Cope, 1868, and the extant Platanista gangetica. The stem cetacean taxon Georgiacetus vogtlensis was used as the outgroup. We removed an undescribed specimen (OU 22125), and added 4 allodelphinid taxa (Zarhinocetus errabundus (Barnes, 2010) Goedertius oregonensis Kimura & Barnes, 2016, Allodelphis pratti and Arktocara yakataga), raising the number of OTUs used in the analysis to 86. We also added codings for four periotic characters (288–291) for Pomatodelphis inaequalis Allen, 1921 and Zarhachis flagellator, based on material available at the Smithsonian Institution (see ‘Specimens observed’). We did not add codings for taxa that we could not directly observe, and therefore some platanistoid taxa were excluded from the analysis, including Huaridelphis raimondii Lambert, Bianucci & Urbina, 2014, Ninjadelphis ujiharai Kimura & Barnes, 2016, and Allodelphis woodburnei Barnes & Reynolds, 2009.

We performed a cladistic search in TNT* (Tree analysis using New Technology) using all characters as unordered and equally weighted. We then conducted subsequent statistical support analyses by searching for successively longer trees to calculate bremer decay indices and 100 bootstrap replicates. The complete matrix in .txt format, as well as a description of character states (Table S1 and Appendix S1) are available in Supplemental Information.

Phylogenetic nomenclature

Following Pyenson et al. (2015), we converted several existing, ranked cetacean taxonomic names into clade names that conform to the principles articulated first by Joyce, Parham & Gauthier (2004). We referred traditional names to more inclusive clades (e.g., in the case of extinct families, node-based clade names), where their composition closely resembles current or widely recognized name applications. For these purposes, we used the abbreviation CCN for Converted Clade Name. Below, we clarify our precise definitions for these clades (see PhyloCode, 2014, Article 9.3; Cantino & De Queiroz, 2014), and we also provide full citations for the names of specifier species.

Nomenclature acts

The electronic edition of this article conforms to the requirements of the amended International Code of Zoological Nomenclature, and hence the new names contained herein are available under that Code from the electronic edition of this article. This published work and the nomenclatural acts it contains have been registered in ZooBank, the online registration system for the ICZN. The ZooBank LSIDs (Life Science Identifiers) can be resolved and the associated information viewed through any standard web browser by appending the LSID to the prefix “http://zoobank.org/”. The LSID for this publication is: urn:lsid:zoobank.org:pub:0194A593-DBE0-47CA-A41F-04A37931BA2F. The electronic edition of this work was published in a journal with an ISSN, and has been archived and is available from the following digital repositories: PubMed Central, and LOCKSS.

Specimens observed

Allodelphis pratti (YPM 13408); Allodelphis sp. (USNM 266608, 256609, 256610); Goedertius oregonensis (LACM 123887); Goedertius sp. (USNM 335406, 335765, 13673, 314421); Notocetus sp. (USNM 206286); Phocageneus venustus (USNM 21039, 475496); Phocageneus sp. (USNM 182939, 362125); Platanista gangetica (USNM 23456); Pomatodelphis bobengi Case, 1934 (299775); Pomatodelphis sp. (USNM 360054); Squalodon calvertensis (USNM 10949, 529246); cast of Waipatia maerewhenua (USNM 508061); Zarhachis flagellator (USNM 299945, 10911, 13768); Zarhachis sp. (USNM 214759, 24868); cast of Zarhinocetus errabundus (USNM 526600); Zarhinocetus errabundus (USNM 11573, 25425).

Results

Systematic paleontology

Cetacea Brisson, 1762	
Odontoceti Flower, 1867 sensu Fordyce & Muizon, 2001	
Platanistoidea (CCN) (node-based version of Fordyce, 1994)	
Allodelphinidae (CCN) (node-based version of Barnes, 2006)	
Arktocara, gen. nov. LSID: urn:lsid:zoobank.org:act:EE11B95B-8338-496B-97F4-1673ED90E709	

Definitions. Crown group Platanista refers to the crown clade arising from the last common ancestor of all lineages descending from Platanista, including two subspecies of Platanista gangetica (P. g. gangetica (Lebeck, 1801) and P. g. minor Owen, 1853), as recognized by The Society for Marine Mammology’ Committee on Taxonomy (2015).

Platanistoidea is a converted clade name for a clade that includes Platanista gangetica and fossil taxa that support familial level taxonomic concepts such as: Allodelphis pratti; Squalodelphis fabianii Dal Piaz, 1917; and Waipatia maerewhenua. We do not formally recognize node-based versions of Squalodelphinidae and Waipatiidae at this time (except for in the Diagnosis section) because these familial level groupings are not the explicit focus of this study, and we defer to future work that can better substantiate their taxonomic scope and better test their monophyly (see, for example, Tanaka & Fordyce, 2014; Tanaka & Fordyce, 2015a). This node-based converted clade of Platanistoidea corresponds to the Fordyce (1994)’s concept of Platanistoidea, but differs from Muizon (1987), and Muizon (1991)’s concept, in its exclusion of Squalodontidae. Following Lambert, Bianucci & Urbina (2014), we exclude Squalodontidae and stem Platanistoidea, such as Prosqualodon davidis Flynn, 1923, and Papahu taitapu Aguirre-Fernández & Fordyce, 2014, from our node-based concept of Platanistoidea. Our concept is more inclusive than Geisler et al. (2011)’s Platanistoidea, which included only Platanista, Zarhachis and Squalodelphis, while excluding Waipatia to outside of crown Odontoceti. Moreover, our concept of Platanistoidea shares very little with Simpson (1945)’s articulation, which included all ‘river dolphin’ lineages, including Inia, Pontoporia, and Lipotes. Subjective synonymies of the converted clade name of Platanistoidea include, among others: Platanistoidea Fordyce, 1994; Platanistoidea Barnes, 2006; Platanistoidea Tanaka & Fordyce, 2014; Platanistoidea Tanaka & Fordyce, 2015a; Platanistoidea Kimura & Barnes, 2016.

Here, we also propose the converted clade name Platanistidae as a node-based clade defined by Platanista, Zarhachis and Pomatodelphis. This node-based converted clade of Platanistidae corresponds to the most recent concepts of the familial level grouping of closest fossil relatives of Platanista, such as Platanistidae Barnes, 2006; Platanistidae Barnes, Kimura & Godfrey, 2010; Platanistidae Geisler et al., 2011; and Platanistidae Bianucci et al., 2013.

Lastly, Allodelphinidae is the converted clade name for the clade that includes the following fossil odontocete genera: Allodelphis, Arktocara, Goedertius, Ninjadelphis, and Zarhinocetus. Subjective synonymies of the converted clade name include: Allodelphinidae Barnes, 2006; Allodelphinidae Lambert, Muizon & Bianucci, 2015; Allodelphinidae Kimura & Barnes, 2016. All previous studies have indicated that Allodelphinidae belongs as a sub-clade within a node-based Platanistoidea.

Type and only included species: Arktocara yakataga, sp. nov.

Etymology. The name Arktocara derives from the combination of arktos from Greek and cara from Latin, which together signify “the face of the North.” The only preserved material of the type specimen, USNM 214830 consists of the cranium, or its face, and its type locality is the furthest north that a platanistoid has ever been found.

Age. Same as that of the species.

Diagnosis. Same as that of the species.

Arktocara yakataga, sp. nov. (Figs. 2–10 and Table 1)

LSID: urn:lsid:zoobank.org:act:FBCF0EAA-7BBB-4EF0-8186-7548993098D1

Figure 7 Skull of the holotype of Allodelphis pratti (YPM 13408) in dorsal view.

(A) Illustrated skull with low opacity mask, interpretive line art, and labels for skull elements. Dotted lines indicate uncertainty of sutures and hatched lines indicate sediment obscuring the fossil. The symbol “?” denotes a displaced skull fragment of unknown origin. (B) photograph of skull in dorsal view, photography by James Di Loreto, Smithsonian Institution. Courtesy of the Division of Vertebrate Paleontology; YPM 13408, Peabody Museum of Natural History, Yale University, New Haven, Connecticut, USA; peabody.yale.edu.

Figure 8 Skull of the holotype of Allodelphis pratti (YPM 13408) in ventral view.

(A) Illustrated skull with low opacity mask, interpretive line art, and labels for skull elements. Dotted lines indicate uncertainty of sutures and hatched lines indicate sediment obscuring the fossil. The symbol “?” denotes a displaced skull fragment of unknown origin. (B) photograph of skull in ventral view, photography by James Di Loreto, Smithsonian Institution. Courtesy of the Division of Vertebrate Paleontology; YPM 13408, Peabody Museum of Natural History, Yale University, New Haven, Connecticut, USA; peabody.yale.edu.

Figure 9 Right periotic of the holotype of Allodelphis pratti (YPM 13408).

Right periotic of Allodelphis pratti in dorsal (A, B), and lateral (C, D) views. (A, C), Illustrated periotic with low opacity mask and interpretive line art. The two periotic synapomorphies for the Platanistoidea are labelled: the parabullary sulcus, and the dorsal crest. (B, D), photography by James Di Loreto, Smithsonian Institution. Courtesy of the Division of Vertebrate Paleontology; YPM 13408, Peabody Museum of Natural History, Yale University, New Haven, Connecticut, USA; peabody.yale.edu.

Figure 10 Strict consensus cladogram with support values.

Phylogenetic analysis of Odontoceti, showing a strict consensus cladogram resulting from 430 most parsimonious trees, 1,960 steps long, with the ensemble consistency index equal to 0.233 and the ensemble retention index equal to 0.631. Numbers below nodes indicate decay index/bootstrap values (bootstrap values <50 were omitted). Labelled circles denode node-based clades, and taxa in the node-based sub-clade of Allodelphinidae are in bold.

Table 1 Measurements for type specimen Arktocara yakataga (USNM 214830).

Measurements of holotype skull of Arktocara yakataga (USNM 214830), in cm (modified after Perrin, 1975 and Tanaka & Fordyce, 2014).

Dimension	Measurement (in cm)	
Total preserved length of skull from furthest anterior point to furthest posterior point	23	
Cranial length from antorbital notches to occipital condyle	17.5	
Distance from preserved rostrum tip to external nares (to mesial end of anterior transverse margin of right naris)	8.5	
Distance between upper margin of foramen magnum and nuchal crest	6.8	
Height of foramen magnum	2.9	
Height of occipital condyle	4.1	
Height of temporal fossa	5.9	
Height of rostrum at base	6.9	
Length of temporal fossa	5.8	
Orbit length	4.9	
Maximum length of nasal fossa of the frontal	2.6	
Length of vertex (nuchal crest to anterior transverse margin of nasal fossa of the frontal)	4	
Depth of rostrum at base	6.5	
Width of rostrum between antorbital notches	8.8	
Width of premaxillae at rostrum base	5.5	
Maximum width of premaxillae on cranium	6.2	
Width of external bony nares	3.6	
Postorbital width of skull	17.7	
Bizygomatic width of skull	19.1	
Width between temporal crests	11.1	
Width of foramen magnum	5.7	
Width of occipital condyles	9.8	

Holotype. USNM 214830, consisting of an incomplete skull lacking the rostrum anterior of the antorbital notches, tympanoperiotics, dentition and mandibles (see Fig. 2).

Type locality. The precise geographic coordinates for the type locality of Arktocara yakataga are unknown. The type specimen (USNM 214830) was discovered and collected in 1951 by United States Geological Survey (USGS) geologist Donald J. Miller (1919–1961), who was mapping what was then the Yakataga District of Alaska (now the Yakutat City and Borough) between 1944 and 1963. Archival notes housed with the specimen at USNM state that Miller found the specimen in the Poul Creek Formation within the then-Yakataga District (see Age, below). Therefore, we delimit the area for the type’s provenance to exposures of the Poul Creek Formation in the Yakutat City and Borough, Alaska, USA, in a grid ranging approximately from 60°22′N, 142°30′W to 60°00′N, 143°22′W (see Fig. 1). While the formation has been named from its exposures along Poul Creek, it has been suggested that the most abundant macrofossils from this unit have been collected from outcrops along Hamilton Creek, White River, and Big River near Reare Glacier (Taliaferro, 1932). It is possible that Miller collected USNM 214830 from one of these exposures.

Formation. Poul Creek Formation.

Age. Archival documentation accessioned in the Department of Paleobiology with USNM 214830 indicate that the type specimen was collected from an unknown locality exposed about 400–500 m below the top of the Poul Creek Formation, which has a total stratigraphic thickness of around 1.9 km (Plafker, 1987). The Yakutat terrane of Southeast Alaska consists of the Kulthieth, Poul Creek, and Yakataga Formations (Perry, Garver & Ridgway, 2009; Plafker, Moore & Winkler, 1994; Miller, 1971). The Kulthieth Formation consists of mostly organic-rich sandstones deposited in nonmarine alluvial, deltaic, barrier beach and shallow marine environments, and is Early Eocene to Early Oligocene (∼54–33 Ma) in age based on the fossil assemblages present (Perry, Garver & Ridgway, 2009). The Upper Eocene to possibly Lower Miocene (∼40–20 Ma) Poul Creek Formation conformably overlies the Kulthieth Formation (Plafker, 1987; Miller, 1971). It is estimated to be approximately 1.9 km thick, and is composed of siltstones and organic-rich sandstones, in part glauconitic recording a marine transgression, interrupted by deposits of the Cenotaph Volcanics (Plafker, 1987). Finally, unconformably overlying the Poul Creek Formation is the Miocene to Pliocene Yakataga Formation (Miller, 1971). It is composed mainly of tillite and marine strata (Perry, Garver & Ridgway, 2009).

The Poul Creek Formation itself is broadly constrained to approximately 40–20 million years in age, from the latest Eocene to possibly early Miocene in age (Plafker, 1987; Miller, 1971). The depositional age of the unit has been further constrained to ∼24 to ∼29 Ma, or a mid to late Oligocene age, based on detrital zircon fission-track analyses of young grain-age populations (Perry, Garver & Ridgway, 2009). Using the broadest time duration for the formation (∼20 million years) and the coarse stratigraphic thickness of the sediments within it (∼2 km), a constant rate of sedimentation would suggest that the stratigraphic position of USNM 214830 at 500 m below the top of the formation would be roughly equivalent to an geochronologic age of ∼25 million years, an estimate that is consistent to the detrital zircon analyses. Overall, we propose a late Oligocene, or Chattian age for Arktocara, although we cannot exclude a Rupelian antiquity.

Diagnosis. Arktocara is a small to medium sized platanistoid odontocete (approximately 2.26 m in total length), which belongs, based on one equivocal synapomorphy, to the node-based Platanistoidea: width: width of the premaxillae >50% of the width of the rostrum at the antorbital notch (character 51[1]). More convincingly, Arktocara belongs to Platanistoidea based on its affinities to other members of the Allodelphinidae that possess unequivocal synapomorphies of the Platanistoidea (see ‘Discussion’ for further comments on the relationship of Allodelphinidae within the Platanistoidea). We also note that, for the purposes of this diagnosis, we use a broad definition of Waipatiidae that includes Otekaikea spp. (see Tanaka & Fordyce (2015a)), and Squalodelphinidae sensu Lambert, Bianucci & Urbina (2014). See ‘Discussion’ for further comments on systematics of these groups.

Arktocara can be differentiated from all other platanistoids by the following combination of character states. Arktocara differs from Waipatiidae and Squalodelphinidae in having: a postglenoid process of squamosal greatly reduced (character 114[1]); an occipital shield bearing a distinct sagittal crest (character 118[1]); length of the zygomatic process as percent of the greatest width of the maxillae across the postorbital processes <30% (character 152[1]); lacking a dorsal condyloid fossa (character 119[0]); and lacking any asymmetry in the vertex (character 98[0]).

Arktocara also differs from Waipatiidae and Platanistidae in having: frontals posterior to the nasals and between the premaxillae wider than the maximum transverse width across the nasals (character 95[0]); and lacking an anterior transverse ridge and large tympanosquamosal recess, with middle sinus still inferred to be present (character 144[1]). Arktocara further differs from Waipatiidae in having: a lacrimal that wraps around the anterior edge of the supraorbital process of frontal and slightly overlies its anterior end (character 37[0]); maxilla forming the dorsolateral edge of the internal opening of the infraorbital foramen (character 43[0]); ventromedial edge of the internal opening of infraorbital foramen formed by maxilla and palatine (character 44[1]); a nuchal crest weakly convex anteriorly in dorsoposterior view (character 117[1]); a lateral end of the groove for the mandibular branch of the trigeminal nerve wrapping laterally around posterior end of pterygoid sinus fossa and opening primarily anteriorly (character 148[0]); the angle formed by the basioccipital crests in ventral view between 15–40° (character 157[1]); and in lacking a premaxillary crest or posterior maxillary crest adjacent to the nasal (character 72[0]).

Arktocara differs from all Platanistidae and Squalodelphinidae in having the anterolateral corner of the maxilla overlying the supraorbital process of frontal being thin and of even thickness to parts posteromedial (character 64[0]). Arktocara also differs from all Platanistidae in having: the apex of the postorbital process of frontal projected posterolaterally and slightly ventrally (character 46[0]); the ratio of the greatest width of the premaxillae to the greatest width of maxillae at the level of postorbital processes between 0.49–0.38 (character 76[1]); a shallow emargination of the posterior edge of zygomatic process by the sternomastoid muscle fossa in lateral view (character 111[1]); the width of the squamosal lateral to the exoccipital in posterior view as a percent of the greatest width of the exoccipitals <15% (character 112[0]); fossae for the preorbital lobe of the pterygoid sinus present in the orbit (character 134[1]); shallow posterior portion of the periotic fossa of the squamosal (character 151[1]); posteroventral-most point on the basioccipital crest forming a closely appressed flange with a narrow crease separating exoccipital dorsally from the rest of crest (character 156[1]); and lacking a pneumatic maxillary crest overhanging medially (character 65[0]). Finally, Arktocara differs from all Squalodelphinidae in having: a narrower width of the premaxillae at the antorbital notches as a percent width of the rostrum at the antorbital notch (50–64%) (character 51[1]).

Arktocara differs from all other Allodelphinidae in having: a reduced postglenoid process of the squamosal (character 114[1]); and the posteroventral-most point of the basioccipital crest forming a closely appressed flange separating the exoccipital dorsally from the rest of the crest by a narrow crease (character 156[1]). Arktocara differs from Allodelphis pratti and Goedertius oregonensis in having: both premaxillae extending posterior to the nasals (character 58[1]); and the ratio of the greatest width of the premaxillae to greatest width of the maxillae at the level of the postorbital processes between 0.49–0.38 (character 76[1]).

Arktocara also differs from Goedertius oregonensis and Zarhinocetus errabundus in having: the width of the premaxillae between 50–64% of the width of the maxillae a the level of the antorbital notches (character 51[1]); a uniformly thin anterolateral corner of the maxilla overlying the supraorbital process of the frontal (character 64[0]); length of the zygomatic process of the squamosal ≤30% of the width of the maxillae at the postorbital processes (character 152[1]); lacking a rostral basin (character 50[0]), lacking a posterior dorsal infraorbital foramen placed posteromedially near the posterior extremity of the premaxilla (character 60[0]); lacking a dorsal condyloid fossa (character 119[0]) and having a symmetrical cranial vertex (character 98[0]).

Arktocara also differs from Allodelphis pratti and Zarhinocetus errabundus in having: two anterior dorsal infraorbital foramina (character 49[1]); and a shallow emargination of the posterior edge of the zygomatic process by the sternomastoid muscle fossa in lateral view (character 111[1]). Arktocara further differs from Goedertius oregonensis having: a fused lacrimal and jugal (character 39[1]); the apex of the postorbital process of the frontal directed posterolaterally and slightly ventrally (character 46[0]); a triangular or anteroposteriorly widened falciform shaped postorbital process of the frontal (character 47[2]); one posterior dorsal infraorbital foramina of the maxilla (character 59[1]); nuchal crest weakly convex anteriorly in dorsoposterior view (character 117[1]); and the posterior edge of the vomer terminating on the basisphenoid (character 154[0]).

Arktocara further differs from Zarhinocetus errabundus in having: the transverse width of the nasal >70% the length of the nasal (character 91[2]); a distinct sagittal crest on the occipital shield (character 118[1]); and in lacking an anterior transverse ridge and large tympanosquamosal recess (character 144[1]).

Lastly, Arktocara displays the following apomorphies: a weakly developed antorbital notch (character 10[0]); straight lateral margin of the right premaxilla posterior to premaxillary foramen (character 56[1]); a U-shaped nasal frontal suture (character 94[2]); and a greatly reduced postglenoid process of the squamosal (character 114[1]).

Etymology. The species epithet ‘yakataga’ derives from the Tlingit name for the point of land along the southeast coast of Alaska between modern day Kayak Island and Ice Bay. This point, currently called Cape Yakataga, is located directly southwest of Watson Peak and represents the southeastern boundary of a floodplain drained by the Bering Glacier. The name Yakataga was first published by Tebenkov (1852: map 7), who was a cartographer and hydrographer of the Imperial Russian Navy, as “M[ys] Yaktaga” on an 1849 map of Alaska. The geographic place name has been alternatively spelled Cape Iaktag, Cape Yakaio, Cape Yakatag, and Yokataga Reef (Orth, 1967). According to the Geographic Names Information System (GNIS, 2016), developed by USGS in cooperation with the United States Board of Geographic Names (BGN), the name “Yakataga” means “canoe road,” referring to two reefs that form a canoe passage to the shore of the village.

Description

Anatomical terminology follows Mead & Fordyce (2009). In most cases, description of individual elements derives from the most informative side of the skull, in terms of preservation; we note any morphological asymmetry if present. Besides overall erosion of the bone surfaces and loss of some skull elements, there appears to be no significant burial-related distortion of the skull.

Skull

The holotype of Arktocara yakataga (USNM 214830) consists of an incomplete skull, measuring 23 cm in preserved length. The majority of the rostrum is missing, with an asymmetric transverse break approximately 2–5 cm anterior to the antorbital notch. The skull also lacks both nasals, jugals, tympanoperiotics, and the right occipital condyle (see Figs. 2 and 3). Most of the paroccipital processes of the exoccipitals are gone, large portions of the palatines and pterygoids are missing, small fragments along the lateral margins of the frontals and maxillae are incomplete, and the general condition of many osteological elements in the skull are poorly preserved. The skull may have been both mechanically and chemically prepared in the past (with no known documentation), including acid preparation, which may have contributed to the poor state of preservation for the osteological surfaces of many elements. Portions of the skull are obscured by a fine-grained grey matrix of siltstone, especially infilling the mesorostral canal, the bony nares, the recesses of tympanoperiotic region, and the braincase (which is exposed via the foramen magnum and fenestrae in the supraoccipital).

In dorsal view, the preserved skull is roughly hexagonal in overall shape (Fig. 2). The external nares are vertically oriented, and positioned at a level between the antorbital and postorbital processes. The vertex is particularly table-like and square, composed of frontals, premaxillae, and nasals (missing). The parietals appear to be narrowly exposed on the vertex immediately anterior to the nuchal crest, between the frontals and supraoccipital—whether this is natural, or an ontogenetic characteristic, is unclear. The vertex is bordered anteriorly by the externals nares, laterally by the maxillae and posteriorly by the nuchal crest of the supraoccipital. The nuchal crest is straight along the posterior edge of the vertex, but begins to curve posterolaterally as it approaches the temporal crest. The temporal fossae are visible in dorsal view due to an intertemporal constriction just anterior to the level of the nuchal crest, and the fossa is floored by a narrow valley (the squamosal fossa) between the squamosal plate and supramastoid crest.

In lateral view, the profile of the skull gradually slopes upwards from the level of the antorbital notch to posterior of the nares, where it levels out on the vertex (Fig. 4). The nuchal crest is well defined, and about the same height as the frontals on the vertex. The orbit is shallowly rounded dorsally (4.9 cm in length), with the maxilla completely overlying the frontal on the thin supraorbital process, except on the postorbital process, where the frontal is exposed laterally. It is unclear whether the antorbital process of the frontal is completely covered by maxilla or not, but most of the medial antorbital process is composed of the lacrimojugal. The temporal fossa is trapezoidal in shape, with the temporal crest forming a right angle with the dorsal margin of the zygomatic process of the squamosal. The dorsal margin of the temporal fossa is roofed over by the frontal.

The preserved posterior portion of rostrum anterior of the antorbital notch is wide (8.8 cm anterior to the antorbital notch) and deep (6.5 cm at the level of the antorbital notch), with a broadly open and deep mesorostral canal (2.4 cm wide and 4.6 cm deep at the level of the antorbital notch). While there is material missing around the antorbital notch, the posterior margin of the right antorbital notch appears to be real, demonstrating it to be weakly developed, forming an L-shape with the lateral margin of the rostrum (Fig. 2). Though some of the anteromedial antorbital process is missing, not enough is missing to have drastically changed the shape of the antorbital notch region. In anterior view, the maxilla abruptly slopes upwards medially to the distinct premaxilla-maxilla suture, and underlies the premaxilla along the entire preserved length of rostrum and likely the cranium (Fig. 5). The premaxilla therefore forms an anteroposteriorly elongated rectangular plateau surrounding the external bony nares, elevated in relation to the plane of the maxillae, appearing similar in transverse cross-section to a horst and graben system.

In ventral view, there is clearly a fossa for the preorbital lobe of the pterygoid sinus (a shallow depression surrounding the ventral infraorbital and sphenopalatine foramina), as well as for the hamular lobe of the pterygoid sinus (fossa anterior to the choanae). Also present are an extremely shallow tympanosquamosal recess for the middle sinus, and a middle pterygoid sinus fossa anterior to the periotic fossa (Fig. 6). There is not enough rostral material or paroccipital process to determine the presence of an anterior or posterior sinus respectively, and the presence of a peribullary sinus is difficult to ascertain.

Premaxilla

Both of the premaxillae are symmetrical, and overlie either the maxillae or the frontals for their entire preserved length (Fig. 2). In lateral view, the premaxilla thins slightly as it passes around the external nares, otherwise maintaining a relatively even thickness on the cranium (Fig. 4). The premaxilla also appears to thin anterior of the antorbital notch (especially in right lateral view), lowering to the same level as the maxilla instead of rising dorsally above it. However, in anterior view, it is evident that the ventral margin of the left premaxilla sinks ventrally into a medial trough created by the maxilla while the dorsal margin remains level, accounting for the apparent reduction in thickness (Fig. 5). Both premaxillae are broken anteriorly, the right further anterior than the left and missing some of its anterolateral edge. The premaxilla-maxilla suture is clear in dorsal view along the entire lateral length of the premaxilla, as well as in anterior view at the transverse cross-section of the rostrum. In dorsal view, the lateral margin of the premaxilla is mostly rectilinear, widening only 0.5 cm from the rostral break to a level anterior to the nares. As it passes laterally around the nares, the premaxilla gently bows out laterally, with the medial edge retreating more than the lateral edge so that the total width is reduced (0.8 cm on the right premaxilla). Posterior to the nares, the lateral edge remains straight posteriorly, but the medial edge expands slightly medially, once again widening the premaxilla. On the left side of the skull, lateral and posterior of the external nares, a narrow ledge of the medial margin of the maxilla laterally borders the premaxilla, where the premaxilla is separated from the maxilla (possibly diagenetically).

On the vertex, the posterior termination of the premaxilla lies on the frontal as an asymmetrical, spatulate lobe, tapering posterolaterally and bordered medially by frontal and the nasal fossa of the frontal, and laterally the maxilla (Fig. 2). There is no division of the premaxilla into a posteromedial splint and posterolateral plate (a widespread characteristic amongst odontocetes). The premaxillary sac fossa, roughly at the level of the antorbital notch, is shallow with a rough surface. No premaxillary foramina or associated premaxillary sulci are visible on the preserved length of the skull.

Maxilla

Only a small portion of rostral maxilla remains, including part of the maxillary flange on the left side, and just over 5 cm of the body of this element on the right side. In anterior view, the maxilla preserved on the rostrum slopes dorsally from the lateral edge to the premaxillary suture (Fig. 5). The premaxillary suture is distinct and unfused. The preserved maxillary flange on the left side is thin and flat. In lateral view, the maxilla gradually increases in depth posteriorly until anterior of the nares, where it reduces in dorsoventral depth to a thin plate forming the facial portion and ascending process of the maxilla (Fig. 4). In dorsal view, the maxilla flares out laterally at the level of the antorbital notch, which is shallow and L-shaped. Posterior to the antorbital process is broad and relatively flat. The right side bears two infraorbital foramina: one immediately posteromedial to the antorbital notch, and one at the level of the nares. The left maxilla has three infraorbital foramina, all in a sagittal plane from immediately posteromedial of the antorbital notch to a level anterior of the nares. The posterior-most dorsal infraorbital foramen on either side is the largest foramen, with two sulci each—one directed posterolaterally and the other posteromedially—giving the foramen a v-shaped appearance in dorsal view. In the facial region, the maxilla gradually curves dorsomedially from the supraorbital process to the premaxillary suture, and the facial fossa is essentially flat. The maxilla does not extend to the postorbital process, though the suture with the frontal on the postorbital process is unclear. The ascending process suddenly curves dorsomedially towards the lateral edge of the vertex, and the posteromedial margin of the maxilla curves dorsolaterally to terminate in a sharp triple-point junction with the nuchal, temporal, and orbitotemporal crests.

In ventral view, the hard palate of maxilla curves dorsolaterally from the midline to the lateral edge, where it flattens out on the maxillary flange (Fig. 3). No alveoli are present in the preserved palatal surface of the maxillae. A small gap between the maxillae along the midline of the hard palate reveals a thin ridge of the vomer, approximately 2.6 cm long and a maximum of 1 mm wide. Just anterior to the level of the antorbital notch, a narrow palatine groove of the maxilla for vessels and/or nerves begins approximately 1.5 cm lateral of the midline, and curves posterolaterally to terminate ventral of the sphenopalatine foramen. Midway along the palatine groove is the posterior palatine foramen, and the posterior portion of the groove is partially roofed over by fragments of the palatine. The distal maxillopalatine suture would have been at some point anterior and lateral of the palatine groove, though the palatine is incomplete and is missing this portion. Between the palatine groove and the medial lamina of the pterygoid, the maxilla is overlain by fragments of the palatine, though the sutures between the two are extremely unclear (Figs. 2, 4 and 6). Lateral of the palatine groove, the maxilla overlies the medial process of the lacrimal, and encircles both the ventral infraorbital foramen and the dorsal margin of the sphenopalatine foramen. Posterior of the foramina, the maxilla terminates in an abutment with frontal.

Frontal

In dorsal view, the frontals are mostly covered by the maxillae, with the exception of exposures on the postorbital processes and the vertex (Fig. 2). The postorbital processes are asymmetrical in lateral view, with a larger and more robust right postorbital process pointing ventroposteriorly and reaching within 0.7 cm of the tip of the zygomatic process (Fig. 4). The left postorbital process is shorter and more smoothly curved ventroposteriorly. Though more of the length appears to be preserved in the right postorbital process than the left, it is difficult to determine whether the asymmetry is real or preservational. Moreover, the dorsal rim of the right orbit is missing and heavily eroded into the supraorbital process, yielding an incomplete view of the orbit on this side of the skull. The frontal-maxillary suture is indistinct as it passes in an anteroposterior direction across the postorbital processes.

The frontal-maxillary suture is also indistinct along the lateral edge of the vertex, where the maxilla rises dorsomedially to the edge of the vertex’s tabular surface (Fig. 2). The sutures are posteromedially convex in dorsal view, on either side of the vertex, curving from the temporal crest to the posterior premaxilla-maxilla suture. On the vertex, the frontals are exposed as a wide, flat surface. They are separated from supraoccipital posteriorly by a narrow exposure of the parietals which may an artifact of ontogeny. As such, it is possible that the frontals did contact the supraoccipital in older individuals of this taxon. The frontal exposure is bordered by the maxillae laterally, and the premaxillae and mesethmoid anteriorly. Between the premaxillae, the frontals bear a shallow fossa for the missing nasal bones (see “Nasal” for further description).

Lacrimojugal

Most of the medial margin of the antorbital process is composed of the lacrimal. In dorsal view, a narrow margin of the lacrimal emerges from beneath the maxilla as a thin plate along the lateral and anterior edges of the antorbital process (Fig. 2). In ventral view, the medial process of the lacrimal extends posteromedially towards the ventral infraorbital foramen, but is overlapped by the maxilla (Fig. 3). The styliform portion of jugal is mostly missing, save a small piece of the jugal base where it is fused to the lacrimal. The jugal base is transversely wide and anteroposteriorly narrow. The lacrimal is covered posteriorly by the frontal.

Nasal

Though both nasal bones are missing, the frontal bones bear a distinct fossa between the premaxillae on the vertex that indicates where the bones would have been (Fig. 2). The fossa is bordered anteriorly by the mesethmoid and laterally by the premaxillae. Based on the extent of the nasal fossa, the nasals were wider anteriorly than posteriorly (from 2.1 to 1.6 cm), and minimally 1.9 cm in length, but could have been longer if they overhung the bony nares, as seen in the holotype of Allodelphis pratti (Fig. 7) The height of the nasals cannot be estimated, considering that they rise above the frontal to varying degrees in similar taxa such as Allodelphis pratti. All codings for the nasals of Arktocara yakataga in the phylogenetic analysis were made from measurements of this nasal fossa of the frontals, considered the minimum extent of the nasals.

Vomer

At the broken rostral tip in anterior view, the mesorostral canal is deeply v-shaped (Fig. 5). The cross-section reveals the damaged vomer to be extremely thin (<1 mm), and lining all sides of the mesorostral canal. Anterior to the nares, the maxilla is exposed dorsal of the vomer, so that it forms the dorsal edge of the mesorostral canal and the anterior wall of the external bony nares, similar to Tursiops truncatus (Montagu, 1821) (based on USNM 504560). Anterior to the nares, the vomer also curves medially to form the medial wall of the external bony nares, and the lateral walls of the nasal septum. Posterior of the nares, the vomer is obscured in dorsal view by unprepared matrix.

In ventral view, the vomer is visible as a long, thin crest running down the midline of the hard palate (Fig. 3). Anterior to the antorbital notch, the vomer is mostly obscured by the maxillae, but emerges at the level of the antorbital notch as a vertical wall separating the choanae, slightly wider at its base and thinning to a sharp crest ventrally. Posterior of the nares, the crest flattens and the vomer flares out laterally, adhering to the basisphenoid. Its suture with the basisphenoid is indistinct, roughly straight transversely, between the posterior lamina of the pterygoid and immediately anterior to the basisphenoid-basioccipital suture.

Mesethmoid

The mesethmoid composes the bulk of the nasal septum, narrowing dorsally and flanked on either side by the thin lamina of the vomer. Behind each choana is a rounded bony protuberance, likely composed of the lateral wings of the mesethmoid (referred to as the ectethmoid by some authors, e.g., Mead & Fordyce, 2009) (Fig. 2).

Parietal

The parietals are visible in dorsal and lateral view in the temporal fossa, where they are in contact with the frontals under the temporal crest, the supraoccipital along the occipital border, and partially underlie the squamosal plate with a semi-circular suture across the temporal fossa. All of the sutures are indistinct. The parietal forms the lateral wall of the braincase as a thin, laterally convex plate. As with the supraoccipital, both parietals in the temporal fossa are broken, with a small rounded window revealing the matrix-filled braincase (Fig. 4). As previously mentioned, the parietals in dorsal view are exposed as a anteroposteriorly narrow strip along the posterior margin of the vertex. In ventral view, the parietal is again visible in the periotic fossa; having passed under the squamosal to form the lateral wall of the braincase, it emerges medial to the squamosal in ventral view as small, slightly concave surface, just posterior to the foramen ovale (Figs. 3 and 6).

Supraoccipital

The supraoccipital is broadly visible in dorsal view, contributing to the entire length of the nuchal crest. The supraoccipital and frontals are very slightly separated along the vertex by an exposure of the parietals sutured to the supraoccipitals. The supraoccipital is also in contact with the parietals along the occipital border (Fig. 2). Lateral of the vertex along the nuchal crest, the supraoccipital is in direct contact with the maxilla. The nuchal crest is straight medially, but begins to curve posteriorly as it approaches the temporal crest. Along the parietal margin, the supraoccipital is a thin plate, with the edge oriented posterolaterally as it curves around the posterior edge of the temporal fossa. In posterior view, the supraoccipital is rectangular in shape (Fig. 5). A prominent external occipital crest divides supraoccipital sagittally, from the midpoint of the nuchal crest to the opisthion. On either side of the crest, the supraoccipital is very thin and slightly concave. Both these surfaces are broken into fenestrae, with rounded margins that reveal openings filled with matrix in the braincase. There is no evidence of a dorsal condyloid fossa of any significance. The contact of the supraoccipital with the exoccipital is indistinct, except around the foramen magnum, where the nuchal tubercle of the supraoccipital clearly tucks underneath the dorsal portion of the occipital condyle.

Exoccipital

Both exoccipitals are incomplete, missing all or part of the occipital condyle, and most of their ventral portions (Fig. 5). The supraoccipital suture is indistinct, but the contact with the squamosal is clear, along the posteroventral temporal crest, and on the ventral side of the skull. The exoccipital is thin along the lateral margin posterior of the temporal crest, thickening ventrally. The broken remains of the occipital condyles are sufficient to observe their robust size and width, composing approximately 70% of the total combined width of the exoccipitals. Only the dorsal portion of the left condyle remains. Its surface is smooth, posteriorly curved and laterally broad. The foramen magnum is elliptical in shape, almost twice as wide as it is tall (2.9 cm wide, 5.8 cm tall). Both ventral and dorsal condyloid fossa are very shallow and ill-defined. Though both exoccipitals are missing most of their ventral portions, including the jugular notches and paroccipital processes, the left exoccipital does bear a small foramen that may represent the hypoglossal foramen, immediately lateral to the posterior end of the basioccipital crest. The exoccipital also forms a small knob separated from the posterior end of the basioccipital crest by a narrow cleft, possibly a vestigial medial crest of the exoccipital.

Basioccipital

In ventral view, the basioccipital widens posteriorly from 6.2 cm wide at its suture with the basisphenoid, to 8 cm at the posterior end of the basioccipital crest (Fig. 3). The element is ventrally concave, with basioccipital crests oriented laterally from a sagittal plane, and at an angle of approximately 12 degrees from the midline (opening posteriorly). The tympanic plates are thin where they overlie the basisphenoid, increasing in width posteriorly before tapering slightly and rounding off at their posterior ends. The right side of the posterior basioccipital crest is missing, though the left side is complete. The posterior end of the basioccipital crest is interrupted by a narrow cleft that separates a small knob from the rest of the tympanic plate. This small knob is immediately medial to the hypoglossal foramen (on the left side), and presumably the jugular notch. Though consistent with the rest of the basioccipital crest, the knob is actually formed by the exoccipital—a condition described in Fordyce’s (1994) description of Waipatia maerewhenua, and also seen in a number of other platanistoids, including Allodelphis pratti (Figs. 7–9). The suture with the basisphenoid, along the anterior margin of the basioccipital, is represented by a wavy margin near the midline. The suture is increasingly less distinct laterally, where the basioccipital extends anteriorly, overlapping the lateral margins of the basisphenoid and bordering the posterior lamina of the pterygoid along its medial edge. There is no strong evidence of a muscular tubercle for the insertion of the ventral rectus capitis muscle.

Sphenoid

The basisphenoid is visible on the ventral side of the skull, though it is mostly obscured by the basioccipital and vomer (Fig. 3). The basioccipital crests extend anteriorly to cover the lateral portions of the basisphenoid, completely obscuring any view of the contact between the basisphenoid and alisphenoid. The posteroventral plate of the vomer obscures the anterior margin of the basisphenoid, and spreads over the basisphenoid’s medial section to terminate immediately anterior to the wavy basioccipital suture, which appears partially fused. In ventrolateral view, the sphenoid re-emerges from beneath the basioccipital, with the ventral carotid foramen tucked under the dorsolateral margin of the basioccipital crest (Fig. 6). A small portion of the basisphenoid is visible, wrapped laterally around the ventral carotid foramen. Anterolateral of the foramen, the alisphenoid extends laterally anterior to the periotic fossa as a thin plate. The alisphenoid passes anterior to the foramen ovale, and bears a long, thin groove for the mandibular nerve, extending anteriorly from the foramen ovale to the anterior margin of the alisphenoid, medial of the subtemporal crest of the squamosal. The alisphenoid continues to extend lateral of this groove, posteriorly contacting the medial edge of the falciform process, and anteriorly fusing to a remnant of the lateral lamina of the pterygoid (see ‘Pterygoid’ below). A very small portion of the orbitosphenoid is visible on the right ventral side of the skull, emerging ventral to the posterior lamina of the pterygoid on the medial wall of the orbit and dorsal of the alisphenoid (only visible on the right side of the skull).

Pterygoid

Both pterygoids are incomplete, missing the pterygoid hamulus and almost all of the lateral lamina. In ventral view, the medial lamina is an extremely thin sheet, meeting the vomer/palatine anterior to the nares and curving posterolaterally to form the anterior and lateral walls of the external bony nares, slightly overlying the palatine (Fig. 3). The posterior lamina rises ventrolaterally as a thin plate, ventrally concave, and forming the posterior wall of the external bony nares. The posterior lamina is bordered posteromedially by the anterior basioccipital crest, separately by a widely open suture. This open suture is unusual among fossil and living odontocetes, and may represent either an ontogenetic feature or diagnostic feature for Arktocara. The posterior lamina is bordered posterolaterally by the alisphenoid.

Platanistoids bear a bony structure on the ventral side of their skulls: a thin, bony lamina that extends from the ventral surface of the hard palate and runs parallel to the posterior lateral lamina of the pterygoid to finally attach to the alisphenoid medial of the squamosal in the ear region. The holotype of Arktocara yakataga is mainly missing this feature, except for a process protruding from the alisphenoid medial of the subtemporal crest. We interpret this feature as an eroded remnant of the lateral lamina of the pterygoid, that is fused to the alisphenoid (there is no detectable suture line between the two elements in Arktocara). For further discussion of the lateral lamina of pterygoid as a platanistoid feature, see discussion in “Platanistoid systematics.”

Palatine

Both of the palatine bones are missing large portions, including the palatine surface, maxillary process, palatal crest, and lateral lamina. However, in ventral view a prominent pterygoid sinus fossa for the hamular lobe is formed by the palatine, anterior to the medial pterygoid lamina and posterior to the palatine groove of the maxilla (Fig. 3). These grooves represent the minimum extent of the palatine extending anteriorly over the maxilla, since the maxillopalatine suture is impossible to detect. A very small fragment of the horizontal portion of the palatine remains immediately ventral to the ventral infraorbital foramen, partially covered by the medial lamina of the pterygoid.

Squamosal

In dorsal view, the short, wide, and rounded zygomatic process of the squamosal points anterolaterally (Fig. 2). The temporal fossa is floored by the squamosal fossa, forming narrow valley between the supramastoid crest and the squamosal plate. In lateral view, the squamosal plate is a thin sheet, slightly convex laterally, and overlaps the parietal at an indistinct, rounded suture traversing the temporal fossa (Fig. 4). The zygomatic process is rounded off. The postglenoid process is greatly reduced and missing its ventral edge on the right side, the postglenoid notch is either absent or too greatly reduced to determine, and the external auditory meatus is difficult to determine. In posterior view, the squamosal is widely visible lateral to the exoccipital, and the temporal crest where these later two elements meet is well developed (Fig. 5). In ventral view, glenoid fossa on the zygomatic process is broad and shallow (Fig. 2). The incomplete postglenoid process is square in cross section. The contribution of the squamosal to the periotic fossa is wide and shallow, sloping medially from the anterior meatal crest, and bordered medially by the parietal exposure in the periotic fossa. The falciform process is transversely thin and flat, and projects ventromedially from the shallow tympanosquamosal recess. Anteromedial of the falciform process, the anterior margin of the squamosal plate and the anterodorsal margin of the falciform process are extended and join to form an anterior protrusion, which overlies the lateral margin of the alisphenoid. This anterior protrusion is observed in other platanistoids, including Platanista and an undescribed platanistoid USNM 214911 (Fig. S1), where it articulates with the alisphenoid and the lateral lamina of the pterygoid Further study will determine if this feature is present in more platanistoid taxa.

Body size estimate

Total body length (TL) was estimated using the formula created by Pyenson & Sponberg (2011) for calculating body size in stem Platanistoidea (sensu Pyenson & Sponberg, 2011) based on a bizygomatic width (BIZYG): LogL=0.92∗logBIZY G−1.51+2.49.

The bizygomatic width of USNM 214930 was measured as 19.1 cm, and using the formula produced a reconstructed body length of 2.28 m. Based on this estimate, Arktocara would have been similar to the adult size of Platanista, which averages a length of 2.4 m and at least 85 kg in weight (Jefferson, Webber & Pitman, 2008). It is likely that, in life, Arktocara possessed a rostrum that was relatively elongate, based on its near relatives Zarhinocetus and Goedertius; the rostrum of Allodelphis is poorly known, based on several incomplete fragments belonging to the type specimen YPM 13408. In addition, all known allodelphinids have elongated necks, indicated by unfused, proportionally large cervical vertebrae with anteroposteriorly elongated centra, which is in contrast to the general cetacean trend of shortened, hydrodynamic necks (Buchholtz, 2001). Though the type specimen of Arktocara lacks any post-cranial material, it most likely also had an elongated neck given its close relationships to other allodelphinids. Such longirostry and neck elongation may add to Arktocara’s reconstructed total length, and although Pyenson & Sponberg’s (2011) equations took such allometry into account, we propose that a TL of 2.28 m for Arktocara may be a slight underestimate.

Ontogeny

We assessed skeletal maturity based on long-established osteological indicators, particularly the fusion of cranial sutures and textural surface of the occipital condyles (Pyenson & Sponberg, 2011). Most sutures are clearly distinguishable and closed, with some exception of sutures on the ventral side of the skull that appear un-sutured. Most pronounced are the open sutures between the dorsal lamina of the pterygoids and the basioccipital on the medial ventral surface (Fig. 3). It is unclear whether this feature is an ontogenetic trait unique to Arktocara, or whether it is more broadly observed in other allodelphinids (for example, Zarhinocetus). Also, the missing nasals and palatines suggest that their sutures to adjacent skeletal elements were unfused. Pyenson & Sponberg (2011) described the presence of a pitted periosteal surface of the occipital condyles as an indication of immaturity. The preserved occipital condyles of USNM 214830 are smooth, indicated a more advanced ontogenetic age. Based on these combined observations, we suggest that the skull of USNM 214930 belonged to a skeletally mature individual.

Phylogenetic analysis results

The phylogenetic analysis resulted in 430 most parsimonious trees, all with a score of 1960, consistency index of 0.233 and retention index of 0.631. The strict consensus tree, which was created from the 430 trees, shows a similar topology to the equally weighted analysis of Tanaka & Fordyce (2015a). Arktocara is the sister taxon to Allodelphis, nested within a broader clade of Allodelphinidae, which includes Zarhinocetus and Goedertius. This is the first phylogenetic analysis to include these latter two genera, which were not included in Barnes (2006)’s original matrix. Lambert, Bianucci & Urbina (2014) recovered a monophyletic Allodelphinidae in their 50% majority consensus tree and Lambert, Muizon & Bianucci (2015) resolved a polyphyletic Allodelphinidae, although both analyses only included Zarhinocetus and Allodelphis among their allodelphinid sample of Platanistoidea. Our analysis yields robust support for the monophyly of Allodelphinidae, with higher support values (decay index 5, bootstrap 50) than those recovered for the node-based clade of Platanistoidea (decay index 1, bootstrap value <50) (Fig. 10). Like Tanaka & Fordyce (2014) and Tanaka & Fordyce (2015a), we failed to recover a monophyletic Squalodelphinidae (sensu Lambert, Bianucci & Urbina, 2014), yet in contrast, we did find low support for a monophyletic Waipatiidae, an idea proposed by Fordyce (1994), but not explicitly tested until recently. Our analysis is not the first one to recover a clade that includes both species of Waipatia and both species of Otekaikea—Tanaka & Fordyce’s (2015a; 2015b) results are consistent with ours—however Tanaka and Fordyce excluded the Otekaikea species from their definition of Waipatiidae. We recommend defining Waipatiidae more inclusively (both species of Waipatia and Otekaikea).

Though it is beyond the scope of this study, the incorporation of molecular data into future analyses of odontocete systematics will help to better resolve the relationships among extant cetaceans, and consequently their fossil relatives. See below for further comment on the implications of these results on the systematics of Platanistoidea.

Discussion

Platanistoid systematics

The present day concept of Platanistoidea has its origins with Simpson (1945), although by the late 20th century, it became clear that genera such as Inia, Pontoporia, and Lipotes were more closely related to Delphinoidea than to Platanista (Muizon, 1984; Muizon, 1985; Muizon, 1987), especially with the advent of molecular datasets in the 21st century (see Geisler et al., 2011 for a comprehensive review). Muizon (1984) provided the first modern articulation of Platanistoidea to include the numerous fossil forms that appeared to be most closely related to Platanista than any other odontocete, living or extinct, including Platanistidae, Squalodelphinidae (=Squalodelphidae sensu Muizon, 1984, an alternative spelling that according to Rice (1998) was wrongly formed), and Squalodontidae. Later, Muizon (1987) described two synapomorphies for Platanistoidea: a loss or reduction of the coracoid process and supraspinatus fossa of the scapula; and the acromion process located on the anterior edge of the scapula. In a review of fossil and extant Delphinida, Muizon (1988a) added another extinct family, Dalpiazinidae, to the aggregate of extinct families in Platanistoidea, tentatively placing it as sister group to Squalodontidae within Platanistoidea.

Muizon (1994) modified this diagnosis of the Platanistoidea to include three more characteristics: a deep subcircular fossa located dorsal to the spiny process of the squamosal; a hook-like articular process or rim on the periotic; and the migration of the palatines dorsolaterally, surrounded by the maxilla and pterygoid which partly overlap them. The type and only specimen of Arktocara does not possess any of the elements required to evaluate these synapomorphies, though the palatine is located dorsolaterally (ventral of the sphenopalatine foramina) and is slightly overlapped by the pterygoid (though not the maxilla) (see ‘Description’, palatine).

Muizon (1994) maintained that Dalpiazinidae may be a sister group to Squalodontidae, but admitted that the available material referable to Dalpiazinidae was too fragmentary to evaluate any synapomorphies of Platanistoidea. As a result, Fordyce (1994) excluded Dalpiazinidae from his analysis of Platanistoidea. Dalpiazina ombonii (Muizon, 1988b) (IGUP 26405), which was originally given the genus name Champsodelphis by Longhi (1898), was later reviewed by and given its new genus by Muizon (1988b), and is the only described member of this group. Based on observations by one of us (NDP) of the type specimen, we follow Fordyce (1994) in excluding this taxon from consideration as a platanistoid until a more detailed study can resolve the confusing history of associated material that forms the basis for this taxon (and potential membership of other odontocetes).

In his description of Waipatia maerewhenua, Fordyce (1994) articulated the current concept of Platanistoidea (and largely the basis for the node-based definition used here), which narrowed Muizon’s (1987, 1991) definition to include only the families Squalodontidae, Squalodelphinidae, and Platanistidae, although Fordyce (1994) hinted at possibly platanistoid affinities of other taxa, such as Prosqualodon davidis. Fordyce (1994) also added two synapomorphies: the anterior process of the periotic roughly cylindrical in cross section; and the anterior process smoothly deflected ventrally. Fordyce (1994)’s diagnosis of Platanistoidea also omitted any mention of synapomorphies related to the palatines, and noted that the previous two synapomorphies of the scapula were equivocal, as they are not seen in all platanistoids. The type specimen of Arktocara has no associated tympanoperiotics, but the periotics of both Allodelphis pratti and Zarhinocetus errabundus possess both periotic synapomorphies of the Platanistoidea (Fig. 9).

More recent revisions of the Platanistoidea have supported the exclusion of Squalodontidae, restructuring Platanistoidea to some combination of the families Platanistidae, Allodelphinidae, Squalodelphinidae and Waipatiidae. Lambert, Bianucci & Urbina (2014)’s description of the squalodelphinid Huaridelphis pointed to the inclusion of Platanistidae, Allodelphinidae and Squalodelphinidae in a monophyletic Platanistoidea based on a number of descriptive synapomorphies: deeply grooved rostral suture between the premaxilla and maxilla; elevation of the antorbital region higher than dorsal margin of rostrum base in lateral view; widening of cranium; presence of a deep fossa in orbit roof; vertex distinctly shifted to the left compared with the sagittal plane of the skull; reduction of the ventral exposure of palatine; hamular fossa of the pterygoid sinus extended anteriorly on the palatal surface of rostrum; presence of an articular rim on the periotic; elongation of anterior spine on the tympanic bulla and associated anterolateral convexity; loss of double rooted posterior teeth; and tooth count greater than 25. Of these synapomorphies, Arktocara lacks two: the antorbital region is not higher than the rostrum base, and the vertex is not shifted to the left. Both Squalodon and Waipatia were excluded from Platanistoidea in the results, though a broader sample size may change the relationship between the heterodont and homodont platanistoids.

In contrast to Lambert, Bianucci & Urbina (2014), Tanaka & Fordyce’s (2015a) equally weighted strict consensus recovered a monophyletic Platanistoidea that included both Waipatia maerewhenua and Waipatia hectori (Tanaka & Fordyce, 2015b), both Otekaikea spp., Platanistidae, Squalodelphis fabianii, and Notocetus vanbenedeni (i.e., a paraphyletic Squalodelphinidae). However, in their implied weighting strict consensus, Squalodon was added to Platanistoidea. Allodelphinidae was not included in their analysis. Tanaka & Fordyce (2015a) diagnosed Platanistoidea sensu stricto (i.e., with Squalodon excluded) based on 6 synapomorphies: presence of the posterior dorsal infraorbital foramina of the maxilla (character 59); C-shaped or weakly curved parabullary sulcus (character 169); presence of the articular rim on the periotic (character 186); presence of the anterior spine of the tympanic bulla (character 195); presence of the anterolateral convexity of the tympanic bulla with anterolateral notch (character 196); and presence of the ventral groove (median furrow) of bulla anteriorly (character 212). Tanaka & Fordyce (2015a) also mentioned that character 59 was seen in other odontocete lineages besides the Platanistoidea, and it is the only character that is preserved in Arktocara.

In a broad review of Allodelphinidae, Kimura & Barnes (2016) described three new allodelphinids from the Miocene of western North America and revised the definition of Platanistoidea to include Waipatiidae, Squalodelphinidae, Allodelphinidae, Squalodontidae, and Platanistidae. Kimura & Barnes (2016), however, did not provide a computer-assisted phylogenetic analysis to support their claim about the familial level relationships among platanistoids, pointing instead to a matrix and an analysis in Barnes (2006) that included only two outgroups in a taxon list that exclusively contained presumed platanistoids. More crucially, Kimura & Barnes (2016) did not perform a phylogenetic analysis nor code the character states for the three novel allodelphinid taxa that they described (i.e., Goedertius oregonensis, Ninjadelphis ujiharai, and Zarhinocetus donnamatsonae Kimura & Barnes, 2016).

Our phylogenetic analysis herein addresses some of the shortfalls of previous studies by including type genera belonging to all potential platanistoid families that have been presented in recent phylogenetic analyses (i.e., Lambert, Bianucci & Urbina, 2014; Tanaka & Fordyce, 2015a; Kimura & Barnes, 2016). We resolved a monophyletic Platanistoidea that included Platanistidae, Waipatiidae (Waipatia maerewhenua + Waipatia hectori + Otekaikea marplesi + Otekaikea huata), Allodelphinidae and a polyphyletic Squalodelphinidae. We note that, for Phocageneus venustus, we followed Tanaka & Fordyce (2015a)’s coding, which is primarily based on USNM 21039 (Kellogg, 1957). Lambert, Bianucci & Urbina (2014) provide a valuable discussion of material that has been referred to this taxon. Our analysis departs most sharply from Tanaka & Fordyce (2015a) with the addition of the four allodelphinid genera. Though our recovery of a monophyletic Waipatiidae consisting of all described species of Waipatia and Otekaikea is consistent with Tanaka & Fordyce (2015b), the authors chose to limit their definition of Waipatiidae to both species of Waipatia, differing from the results in Tanaka & Fordyce (2014) where the authors definied Waipatiidae as including Waipatia maerewhenua and Otekaikea marplesi. Our results are consistent with Tanaka & Fordyce (2015a)’s findings with the resolution of a polyphyletic Squalodelphinidae, with Squalodelphis fabianii as a basal member of Platanistoidea and an unnamed clade of Notocetus vanbenedeni + Phocageneus venustus as the sister group to Platanistidae. A more detailed coding of Squalodelphinidae in future work, especially one that includes Huaridelphis raimondii, will provide more insight into the relationships among this group.

We diagnose a node-based Platanistoidea by the following synapomorphies: moderately elevated coronoid process (character 33*); premaxillae >65% of width of rostrum at antorbital notches (character 51*); alisphenoidal-squamosal suture coursing along groove for mandibular branch of trigeminal nerve in ventral view (character 147[1]); lateral groove or depression with profile of periotic becoming slightly to markedly sigmoidal in dorsal view (character 166[1]); anteroposterior ridge on dorsal side anterior process and body of periotic (character 167[1]); parabullary sulcus on the periotic weakly to strongly curved and c-shaped (character 169[1,2]); and ventral surface of the posterior process of the periotic not flat along a straight path perpendicular to its long axis (character 191[1,2]) Of these synapomorphies, the two marked by an asterisk (∗) are equivocal across the group, demonstrating character state reversals or independent origins (characters 33, 51). Two characters are ambiguous and show independent origins (characters 169 and character 191), but we argue remain useful for characterizing this group.

Only one of the six synapomorphies presented by Tanaka & Fordyce (2015a) is consistent with ours (character 169). The other 5 characters are all equivocal across the Platanistoidea, but some are still useful for diagnosing members of certain sub-clades. For example, the presence of the articular rim or on the periotic (character 186) is seen in all platanistoids except Allodelphis pratti, where there is no distinguishable rim lateral to the posterior process and separated by a sulcus (Fig. 9). In Zarhinocetus errabundus, this trait is present as an extremely reduced rim. Kimura & Barnes (2016) make no mention of an articular rim or process on the periotic of Ninjadelphis ujiharai, and there is no evidence of it from the published photos of the type specimen. The presence of the anterior spine of the tympanic bulla (character 195), the anterolateral convexity of the tympanic bulla with anterolateral notch (character 196), and the ventral groove (median furrow) of bulla anteriorly (character 212) are all ambiguous characters, represented by two states each across Platanistoidea. All of the latter traits are present in Allodelphis pratti and Zarhinocetus errabundus, with perhaps the exception of the ventral groove of the anterior surface of the bulla in Allodelphis pratti, which could not be determined from the photos of the referred specimen (UCMP 83791) provided by Kimura & Barnes (2016), nor was not mentioned in their description of this taxon.

Systematics of Allodelphinidae

Our analysis recovered Allodelphinidae as a well-supported sub-clade within a node-based Platanistoidea, rooted in a polytomy with Squalodelphis fabianii and an unnamed sub-clade that includes Notocetus vanbenedeni + Phocageneus venustus + Platanistidae. Allodelphinidae in our study is supported by the following synapomorphies: rostral constriction anterior to the antorbital notch (character 9[1]); premaxillae in dorsal view contacting along midline for approximately half of the entire length of the rostrum and partially fused (character 14[3]); buccal teeth entocingulum absent (character 24[1]); greatest diameter of largest functional tooth <3% of greatest width of maxillae at postorbital processes (character 25[2]); angle of anterior edge of supraorbital process and the median line oriented anteromedially (character 35[1]); dorsolateral edge of internal opening of infraorbital foramen formed by maxilla (character 43[0]); posterolateral sulcus shallow or absent (character 57[1]); lack of premaxillary crest or posterior maxillary crest adjacent to nasals (character 72[0]); temporal fossa roofed over by lateral expansion of the maxillae (character 101[1]); palatines partially covered by pterygoid dividing it into medial and lateral exposures (character 121[1]); lateral lamina of palatine (character 122[1]); lateral end of groove for mandibular branch of trigeminal nerve wrapping laterally around posterior end of pterygoid sinus fossa and opening anteriorly (character 148[0]); lack of anterior bullar facet (character 172[1]); elevated caudal tympanic process of periotic with ventral and posterior edges forming a right angle in medial view (character 178[1]); tubular fundus of internal acoustic meatus (character 182[1]) angle between posterior process of periotic and long axis of pars cochlearis ≤135° from dorsal or ventral view (character 189[1]); and ventral surface of posterior process of periotic convex along a straight path perpendicular to its long axis (character 191[2]). Based on the published descriptions and illustrations provided by Kimura & Barnes (2016), the three allodelphinid taxa not included in our phylogenetic analysis (Allodelphis woodburnei, Ninjadelphis ujiharai, and Zarhinocetus donnamatsonae) each possess all of the allodelphinid synapomorphies presented by our analysis.

In their review of Allodelphinidae, Kimura & Barnes (2016) based their diagnosis of this group on comparative characters rather than phylogenetic synapomorphies. Many of these comparative characters can be readily observed in all platanistoids, such as the posteriorly extended lateral lamina of the pterygoid and palatine (except in species of Waipatia and Otekaikea where the palatine is poorly preserved or missing), and a tympanic bulla with elongated and pointed anterior process, among others. Nevertheless, our diagnosis is consistent with Kimura & Barnes (2016)’s concept of Allodelphinidae with only two exceptions. First, Kimura & Barnes (2016) report that, in allodelphinids, the posterior ends of the premaxillae are separated from the lateral sides of the corresponding nasal bones, beginning with a more “primitive” state in Allodelphis pratti where only one premaxilla is separated from the corresponding nasal by a tiny exposure of maxilla, to further “derived” states in Ninjadelphis and Zarhinocetus where the premaxillae are further retracted anteriorly onto the facial region and away from the nasals. However, it is unclear in the more “primitive” state of Allodelphis whether the lack of contact between the premaxilla and nasal could be a result of diagenetic breakage, or individual variation. Furthermore, speculations on the more “derived states” in taxa such as Ninjadelphis, are based on specimens with incomplete premaxillae. In Goedertius oregonensis, the premaxillae are not separated from the nasals. This condition is likely also true for Arktocara yakataga: although the nasals are missing, the premaxillae directly abut the nasal fossa of the frontal, and therefore would most likely have been in direct contact with the nasals. Further extensive comparative work on allodelphinid taxa (including the multiple specimens housed at USNM that can readily be referred to Goedertius sp. (Fig. S2)) will help to clarify the distribution and diagnostic utility of these traits.

Second, Kimura & Barnes (2016) diagnosed Allodelphinidae by an absence of both the preorbital and postorbital lobe of the pterygoid sinus. Both fossae for the pre- and postorbital lobe of the pterygoid sinus are unclear in the type specimen of Allodelphis pratti, in part due to obstruction by unprepared matrix. However, in Arktocara yakataga, though there is no obvious indication of a postorbital lobe of the pterygoid sinus, the deep and broad fossa surrounding the ventral infraorbital foramen and the sphenopalatine foramen anteromedial of the orbit suggests the presence of a preorbital lobe.

Originally assigned to Platanistidae by Wilson (1935), Allodelphis pratti was referred to the Platanistidae by Barnes (1977), and later Barnes (2006) erected a new group, Allodelphinidae, for it. However, in both instances, Barnes (1977) and Barnes (2006) did not provide an explanation for why the Allodelphinidae belong to the Platanistoidea. Of the 7 synapomorphies for Platanistoidea identified by our phylogenetic analysis, the Allodelphinidae possessed 4 of the 5 unequivocal characters: lateral groove or depression with profile of periotic becoming slightly to markedly sigmoidal in dorsal view (character 166[1]); anteroposterior ridge on dorsal side anterior process and body of periotic (character 167[1]); parabullary sulcus on the periotic weakly to strongly curved and c-shaped (character 169[1,2]); and ventral surface of the posterior process of the periotic not flat along a straight path perpendicular to its long axis (character 191[1,2]). The fifth unequivocal character (147), could not be observed in any of the Allodelphinidae specimens. In addition, the type specimen of Ninjadelphis is the only allodelphinid specimen with an associated scapula, and it is missing the coracoid process. This agreement with Muizon’s (1987) platanistoid synapomorphy—the loss or reduction of the coracoid process of the scapula—suggests that, though the process is still present in some putative platanistoids (i.e., Otekaikea huata), the character may still be relevant for diagnosing Platanistoidea (Kimura & Barnes, 2016). We urge future studies on Allodelphinidae to not only include all available genera (if not putative species), but also to explicitly test phylogenetic hypotheses in a repeatable analytical framework.

Morphological comparisons

Of the 7 supporting synapomorphies for Platanistoidea in our study, none of the unequivocal synapomorphies are preserved and demonstrated on the skull of Arktocara. However, one equivocal synapomorphy is preserved in Arktocara: width of the premaxillae >50% of the width of the rostrum at the antorbital notch (character 51[1]). Though the type specimen of Arktocara lacks tympanoperiotics, it is closely allied with Allodelphis pratti, whose periotic shares three more of the platanistoid synapomorphies: presence of lateral groove or depression with the profile of the periotic becoming slightly to markedly sigmoidal in in dorsal view (character 166[1]); anteroposterior ridge developed on anterior process and body of periotic in dorsal view (character 167[1]); and a curved C-shaped parabullary sulcus (character 169[2]; see Fig. 9 for illustration of the periotic synapomorphies on the type specimen of Allodelphis pratti). Therefore, in the absence of tympanoperiotics associated with new cranial material of Arktocara, we are confident that these elements would share many features with Allodelphis pratti, the sister taxon of Arktocara.

Overall, the allodelphinid that most resembles Arktocara in morphology is Allodelphis pratti (Figs. 7–9), originally described by Wilson (1935) from the Jewett Sand in Kern County, California, USA. Having examined the holotype—consisting of a skull with associated rostrum and jaw fragments, other skull fragments, and a right periotic—we note that the material included in the holotype actually belongs to more than one individual. For example, the holotype includes an isolated left postglenoid process, even though the holotype skull still has this feature intact. However, having examined the right periotic in relation to the periotic fossa, we are confident in the association of this element to the holotype skull. Any further study including the holotype of Allodelphis should take this caveat into consideration. Allodelphis is similar in size and shape to the type of Arktocara, with wide, hexagonally shaped craniums and postorbital widths within 2 cm of one another. In dorsal view, the two genera are alike in having their premaxillae rise above the maxillae for the entire length of the cranium from the level of the antorbital notch to the cranial vertex, forming an anteroposteriorly elongated and dorsally elevated plateau in relation to the broad, flat maxilla across the facial region. In both genera, this premaxillary plateau continues posteriorly to a tabular vertex, posterior to the external bony nares. The exposures of the frontals and nasals are symmetrical on the vertex, and there is no evident leftward skew or other facial asymmetry. The nasals are also transversely widened anteriorly, setting these two genera apart from all other allodelphinids. Kimura & Barnes (2016) showed photos of the holotype of Allodelphis pratti with a feature on the skull labeled “posterior dorsal infraorbital foramen.” However, having examined the holotype, we do not see any evidence of a posterior infraorbital foramina—the featured labelled in the photo is a small break in the maxilla overlying the frontal. While it is possible that a posterior dorsal infraorbital foramen is hidden under the jaw fragment that is adhered to the right maxilla, we refrain from definitively stating any foramen exists. Both Arktocara yakataga and Allodelphis pratti have a nuchal crest weakly convex anteriorly, a widely open mesorostral canal anterior to the bony nares, the maxilla covering almost all of the frontal along the supraorbital process, and the posterior end of the basioccipital crest separated from the rest of the crest by a narrow crease.

The coded character state differences between Arktocara yakataga and Allodelphis pratti are listed in the Diagnosis section, above, although we provide more descriptive differences between these two taxa, as follows. First, Arktocara differs from Allodelphis in dorsal view by having: a deeper mesorostral canal anterior to the external nares; straight lateral margins of the premaxillae lateral and posterior of the external nares; no exposure of the maxillae on the vertex; a greater transverse constriction of the lateral margins of the maxilla/frontal anterior to the nuchal crest; a less extreme flaring of the posterior temporal crest along the occipital border; and more prominent dorsal infraorbital foramina, with posteriorly directed sulci. In lateral view, Arktocara shows a markedly reduced postglenoid process and zygomatic process of the squamosal, and a more posterolaterally directed postorbital process as opposed to a ventrally oriented process in Allodelphis. In ventral view, Arktocara has a more elevated vomerine keel. We argue that these differences, along with those coded in the phylogenetic analysis, provide the basis for Arktocara yakataga’s status as a new genus of allodelphinid.

Arktocara also differs in clear ways from three allodelphinids (sensu Kimura & Barnes, 2016) that were not included in the phylogenetic analysis: Ninjadelphis ujiharai, Allodelphis woodburnei, and Zarhinocetus donnamatsonae. Arktocara differs from both Ninjadelphis ujiharai and Zarhinocetus donnamatsonae in having: a wider opening of the mesorostral canal, anterior to the external nares in dorsal view; anteroposteriorly straight lateral margins of the premaxillae both lateral and posterior of the external bony nares, in dorsal view; the posterior ends of the premaxillae extending posterior of the nasals; nasals expanding in width anteriorly rather than narrowing anteriorly; a reduced post-glenoid process; and a broader extent of the maxilla above the supraorbital process of the frontal. Arktocara further differs from both Ninjadelphis ujiharai and Zarhinocetus donnamatsonae in lacking a dorsal depression on the base of the rostrum formed by ventromedially sloping of the premaxillae and maxillae, and lacking an asymmetrical skew to the vertex or nuchal crest.

Arktocara further differs from Ninjadelphis ujiharai in: lacking exposures of the maxillae on the vertex; lacking a glenoid fossa facing anteriorly as opposed to anteromedially; lacking widely diverging basioccipital crests; and lacking a depressed pit of the posterior end of the maxilla with an overhanging lip of the nuchal crest. Arktocara also differs from Zarhinocetus donnamatsonae in having: a more prominent and flaring temporal crest; a zygomatic process more tapered anteriorly in lateral view; the absence of a maxillary tuberosity on the lateral edge of the maxillary flange immediately anterior to the antorbital notch; no reduction of the maxilla on the supraorbital process to expose a thick band of frontal; and lacking a maxillary crest on the supraorbital process in dorsal view. Arktocara differs from Allodelphis woodburnei in having: a smaller and more anteriorly tapered zygomatic process; a reduced postglenoid process; the absence of a prominent fossa on each side of the sagittal crest on the supraoccipital; the premaxillae sloping medially towards the mesorostral canal on the posterior rostrum; and a glenoid fossa directed anteriorly rather than anteroventrally.

Geological & geographic significance

Today, Platanista gangetica is distributed in two subspecies across the Indus, Ganges-Brahmaputra-Megna and Karnaphuli-Sangu river systems of Southeast Asia, and remains highly threatened by human activities, including by-catch, fishing, and habitat modification (e.g., Braulik et al., 2014a). The fossil record of all other Platanistoidea demonstrates that the immediate relatives of Platanista gangetica comprise a morphologically diverse group of small to medium sized odontocetes that are distributed globally in marine sediments of Oligocene and Miocene age (see Bianucci et al. (2013) and Hulbert & Whitmore Jr (2006) for two exceptional occurrences of platanistid specimens in freshwater sediments of Peru and Alabama, respectively). There is no fossil record for the genus Platanista, but recent work on mitochondrial DNA haplotype diversity (Braulik et al., 2014b) places the divergence between subspecies across at around 550,000 years ago (with 95% posterior probability 0.13–1.05 million years ago). The strong ecological disparity between Platanista’s obligate freshwater lifestyle and the presumed marine lifestyle of all other named platanistoids (Fig. 11) implies some kind of differential evolutionary success for this group, with potentially higher extinction rates in Platanistoidea. Fordyce & Muizon (2001) first proposed that competition between platanistoids and early delphinioids may explain the strong difference in taxonomic richness observed in their fossil records, but this suggestion has never been tested in a rigorous framework (Fordyce, 2003; Marx, Lambert & Uhen, 2016).

Figure 11 Phylogenetic results of Platanistoidea and major odontocete groups, calibrated for geologic time.

Time calibrated phylogenetic tree of the Platanistoidea, pruned from the consensus cladogram in Fig. 10. The groups ‘stem Odontoceti’ and ‘all other Odontoceti’ were left as collapsed outgroups. Stratigraphic range data were derived from published accounts for each taxon, including global ranges. Geologic time scale based on Cohen et al. (2013). Stem Odontoceti node depth follows mean divergence date estimates by McGowen, Spaulding & Gatesy (2009); all other nodes (Platanistoidea, Allodelphinidae) should be considered graphical heuristics, and do not reflect divergence dates. Thick bars correspond to the stratigraphic ranges of each taxon, with arrows indicating lower confidence in stratigraphic boundaries. Ecological habitat preference (freshwater vs. marine) is indicated by bar colour, and is based on depositional environment or extant habitat. Labelled circles denode node-based clades. Abbreviations: Aquitan., Aquitanian; H., Holocene; Langh., Langhian; Mess., Messinian; P., Piacenzian; Ple., Pleistocene; Plioc., Pliocene; Serra., Serravallian; Zan., Zanclean.

Platanistoids first appear in the fossil record in the late Oligocene, and reach peak richness in the early Miocene (Kimura & Barnes, 2016; Tanaka & Fordyce, 2015a). The oldest platanistoids with solid age constraints are the waipatiids, all found in the Oligocene-Miocene Otekaike Limestone of New Zealand (Graham et al., 2000; Benham, 1935; Fordyce, 1994; Tanaka & Fordyce, 2014; Tanaka & Fordyce, 2015a). Based on both the lithology and the presence of age-diagnostic planktic foraminifera and ostracod species, Waipatia hectori (Benham, 1935) is the oldest reported waipatiid, from the uppermost Duntroonian Stage of the Otekaike Limestone, approximately 25.2 Ma (Tanaka & Fordyce, 2015b). Arktocara is possibly very similar in age to Waipatia hectori, constrained to the Chattian Stage of the upper Oligocene in the Poul Creek Formation, approximately ∼24–29 Ma (Perry, Garver & Ridgway, 2009). Unfortunately, the lack of robust locality data for either Waipatia hectori or Arktocara makes impossible to determine which is the oldest.

Arktocara is, however, very clearly the oldest known allodelphinid, expanding the previously reported age range of Allodelphinidae by as much as 9 million years (Kimura & Barnes, 2016). Other allodelphinids span temporally from the early to middle Miocene, which largely matches the stratigraphic range of other platanistoid lineages (Fig. 11). Interestingly, Arktocara is among the oldest crown Odontoceti, reinforcing the long-standing view that the timing for the diversification for crown lineages must have occurred no later than the early Oligocene.

Lastly, Allodelphinidae appear uniquely limited, in terms of geography, to marine rocks of the North Pacific Ocean, with occurrences in Japan, Alaska, Washington State, Oregon, and California (see Fig. 12; Kimura & Barnes, 2016). Arktocara expands this geographic range to sub-Arctic latitudes. At approximately 60°N in the Yakutat City and Borough, Arktocara is the most northern platanistoid yet reported. The next most northern platanistoid reported is an incomplete and unnamed specimen from the late Chattian marine Vejle Fjord Formation in northern Denmark, approximately 56.7°N, 9.0°E (Hoch, 2000).

Figure 12 Distribution map of fossil Allodelphinidae.

Mapped of fossil localities of allodelphinids, projected on a truncated Winkel Tripel map and centered on 25°N and 170°W. Occurrences for fossil data derive from location of type and referred localities for each taxon, are listed alphabetically by region, and are represented by orange dots.

Supplemental Information

Figure S1 Morphology of the squamosal in platanistoids

A homologous feature of the squamosal is observed in multiple platanistoid taxa: (A) Arktocara yakataga (USNM 214830), (B) Platanista gangetica (USNM 23456) and (C) USNM 214911, an undescribed platanistoid. This feature (highlighted in red), is a pointed lamina projected anteromedially from the anterior margin of the falciform process (highlighted in blue), posterior to the lateral lamina of the pterygoid and anterior to the posterolateral lamina of the alisphenoid that articulates with the base of the falciform process (see Figs. 3 and 6 for additional reference). Arrows indicate anatomical direction, with a, anterior, l, left lateral, and r, right lateral.

Click here for additional data file.

Figure S2 Referred specimens of Goedertius sp

Photographs of undescribed platanistoid specimens housed in the Vertebrate Paleontology collections of the National Museum of Natural History, Smithsonian Institution, Washington D.C. All of the skulls are referred in this paper to the allodelphinid genus Goedertius. (A) USNM 335406, (B) USNM 335765, (C) USNM 314421, (D) USNM 13673.

Click here for additional data file.

Table S1 Matrix constructed in Mesquite for Odontoceti including Arktocara yakataga, .txt format

0, primitive state; 1, 2, 3, derived states; 0/1, a variable between 0 and 1; 1/2 a variable between 1 and 2; 1/3, a variable between 1 and 3; ?, missing character or taxon not coded for this character. Following Tanaka & Fordyce (2015a), with the removal of the undescribed specimen OU 22125, and addition 4 allodelphinidae taxa (Zarhinocetus errabundus, Goedertius oregonensis, Allodelphis pratti and Arktocara yakataga). Also added were codings for four periotic characters (288–291) for Pomatodelphis inaequalis and Zarhachis flagellator.

Click here for additional data file.

Appendix S1 Character state descriptions

Following Tanaka & Fordyce (2015a). Modifications new to this study are noted.

Click here for additional data file.

We are indebted to LL Jacobs, JF Parham, J Velez-Juarbe, DJ Bohaska, MR McCurry and CM Peredo, who all provided valuable historical insights and warm encouragement for the description of the type specimen herein. Many thanks also to M Miller, son of DJ Miller, whose correspondence with LL Jacobs provided knowledge of his father’s career. We are grateful for the support of the NMNH Imaging, namely J Di Loreto, who provided the beautiful photography of the type specimen Arktocara yakataga and the type of Allodelphis pratti. We also thank to the Division of Vertebrate Paleontology at YPM, especially C Norris, D Brinkman, and M Fox, for coordinating the loan of the type specimen of Allodelphis pratti for comparative purposes. We appreciate the logistical support of DJ Bohaska, M Pinsdorf, P Kroehler, S Jabo, and MT Carrano. We also thank H Little, J Blundell, A Metallo, V Rossi, and the Smithsonian Institution’s Digitization Program Office 3D Lab, for technical advice, expertise, and access to equipment. Chesapeake Testing (Belcamp, Maryland) provided access and resources for microCT scanning, and we thank C Peitsch, R Peitsch, and C Schueler for help. Materialize NV (Leuven, Belgium) provided helpful technical support with 3D model rendering. We also thank J Velez-Juarbe, who provided crucial discussion and assistance, as well as access to valuable unpublished data. Lastly, we are very grateful to JF Parham, who shared his expertise and invaluable guidance in regards to phylogenetic nomenclature.

Anatomical Abbreviations

al. alisphenoid

bsph. basisphenoid

ex. exoccipital

falc. process falciform process

f. ovale foramen ovale

fr. frontal

lac. lacrimojugal

ll./ lat. lam. pterygoid lateral lamina of the pterygoid

infr. Foram. infraorbital foramina

Ma. mega-annum period of 1 million years

max. maxilla

m. lam. pt. medial lamina of the pterygoid

mes. mesethmoid

ns. nasal

o./ orb. orbitosphenoid

pal. palatine

par. parietal

p. glenoid process postglenoid process

p. lam. pterygoid posterior lamina of the pterygoid

pmx. premaxilla

pt. pterygoid

sphpal. foram. sphenopalatine foramen

v. ventral

? displaced skull fragment of unknown origin

Institutional Abbreviations

IGUP Geological Institute of Padua University, Padua, Italy.

LACM Departments of Mammalogy and Vertebrate Paleontology, Natural History Museum of Los Angeles County, Los Angeles, California, USA.

OU Geology Museum, University of Otago, Dunedin, New Zealand.

UCMP University of California Museum of Paleontology, Berkeley, California, USA.

USNM Departments of Paleobiology and Vertebrate Zoology (Division of Mammals), National Museum of Natural History, Smithsonian Institution, Washington, District of Columbia, USA.

YPM Division of Vertebrate Paleontology, Yale Peabody Museum, New Haven, Connecticut, USA.

Additional Information and Declarations

Competing Interests

Author Contributions

Data Availiability

New Species Registration

Nicholas D. Pyenson is an Academic Editor for PeerJ. This does not alter the authors’ adherence to PeerJ policies on sharing data and materials.

Alexandra T. Boersma and Nicholas D. Pyenson conceived and designed the experiments, performed the experiments, analyzed the data, contributed reagents/materials/analysis tools, wrote the paper, prepared figures and/or tables, reviewed drafts of the paper.

The following information was supplied regarding data availability:

Zenodo: DOI: 10.5281/zenodo.51363.

The following information was supplied regarding the registration of a newly described species:

The following information was supplied regarding the registration of a newly described species:

Publication LSID: urn:lsid:zoobank.org:pub:0194A593-DBE0-47CA-A41F- 04A37931BA2F

Arktocara LSID: urn:lsid:zoobank.org:act:EE11B95B-8338-496B-97F4-1673ED90E709

Arktocara yakataga LSID: urn:lsid:zoobank.org:act:FBCF0EAA-7BBB-4EF0-8186-7548993098D1.

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
