# Peer review of "Arktocara yakataga, a new fossil odontocete (Mammalia, Cetacea) from the Oligocene of Alaska and the antiquity of Platanistoidea"

_PeerJ, doi:10.7717/peerj.2321_

## Round 0.1 · original submission · Minor Revisions

Three reviewers have provided extensive minor suggestions for your manuscript, I believe that amending your manuscript accordingly will improve it. If you consider some of their comments as not-valid and decide not to make some changes, please provide me with an explanation thereof.

The reviewers also had some more serious comments, reviewer 2 suggested redoing the photographs. While I agree that some could stand some increase in contrast, I do not believe it is necessary to retake them. Reviewer 3 suggested including molecular data in the phylogenetic analysis. I believe that adding a sentence that explains why you did not is sufficient.

I have two comments myself. The map that you provided is puzzling. You indicate that it comes from Miller, 1971 and is downloadable from usgs.gov. Please provide a more complete address, I could not find it at that website. However, I do know those rocks some, and note that they are mapped on the (modern www-based) geological map of Alaska (behind the paywall at USGS.gov) as 4-5 different formations of early Tertiary age, what you mark goes much beyond the Poul Creek Formation. The Poul Creek, as originally defined by Tagliaferro, is more narrowly defined than what you indicate. Of course, I am willing to use your definition of the formation if there is published documentation for it or you make an argument as to why these rocks should be included.

I do object to the term Pan-Platanista because I believe that hyphenated terms zoological nomenclature are confusing and complicate computer searches. You also italicize it, suggesting it is a genus name, which it is not (since you list it above the superfamily level). Hyphenated names are (nearly) prohibited for species names by the code, and I believe that at least the hyphen and italics should be removed. However, I will run this by my fellow PeerJ systematically inclined editors, as it would be good if there is consensus within the journal.

·

Basic reporting

Review by O. Lambert

The whole text is clear and easy to follow, with a well-organized structure. The introduction provides a useful background for the analysis of the specimen.
The literature list is enough detailed. Some references from the text were not found in the list (see below) and a few references in the list were not found in the text (see annotated pdf):

Lambert et al. 2015
Marx & Fordyce, 2015
Muizon, 1988 (polyphylétisme des Acrodelphidae)
Uhen & Pyenson, 2007

The figures are of good quality and correctly labeled (see punctual issues mentioned below and in the annotated ms, especially for the need to label some diagnostic characters on the photos of the holotype). I would suggest lightening somewhat the different photos of figure 10.

Experimental design

No comments

Validity of the findings

The definition of this new genus and species is well supported by a large amount of characters (but see comments below on the possible over interpretation of some morphological features). The phylogenetic analysis is convincing. Both the description and the phylogeny support well the conclusions.

Additional comments

Find below comments about some issues. Most are minor, except for some possible problems in the diagnosis and some anatomical interpretations. I also annotated the pdf of the ms with other questions/suggestions for both the text and figures.

l. 69: 'have reached a general consensus about the inclusion...': This is not true for Squalodontidae, as you mention it just below. This family should be removed from the list.

l. 123: 'type (and only known) specimens': Written like this, it looks like these taxa are based on a single specimen. This is not true for Huaridelphis, with several specimens referred. (just a detail)

l. 217: 'yet note that stem Platanistoidea, such as ...': I am not sure to follow you here. In your consensus tree (fig. 11), Prosqualodon, Squalodon and Papahu fall outside stem Platanistoidea. It seems arbitrary to include them in Pan-Platanista in that tree, as they may prove in better-resolved trees to fall either as stem odontocetes or as early members of the non-platanistoid crown odontocete clade. Furthermore, in fig. 12 you modify the topology of the tree, removing all other crown odontocetes and therefore having these three taxa as more closely related to Platanistoidea. I would suggest correcting fig. 12 to keep the same topology as in fig. 11, and modifying the text of lines 217-... to better match the obtained relationships. Otherwise, your statement should be better supported by a discussion.

Diagnosis:
In general, I find it useful to have such a detailed diagnosis. However, when listing characters obtained a posteriori from the cladistic analysis, there is a risk that the diagnosis departs from the description and figures. Indeed, coding of characters generally does not occur simultaneously to the description and preparation of figures.
In some cases I feel that the coding of characters may correspond to some degree of over-interpretation of the preserved elements (i.e., codings for regions that are only poorly preserved and for which a question mark may have been more reasonable). Because I did not see the specimen, I cannot be sure that a structure is indeed too poorly preserved for being coded.
In other cases the characters in the diagnosis seem not to match the description. I suggest labeling a larger amount of diagnostic characters in the corresponding figures. This will better support your codings and diagnosis.
Some examples of possible issues and suggestions here:
l. 297-298: 'alisphenoid-squamosal suture coursing...': Both the suture and the groove for the mandibular branch should be labeled in one of the figures.

l. 301: 'sternomastoid muscle fossa': to be labeled in fig. 4.

l. 313: 'almost straight in lateral view': The ventral edge of the zygomatic process seems to appear strongly concave on both the right and left sides. It would also help if you labeled the postglenoid process in lateral view.

l. 314: 'fossa for inferior vestibule': This structure is neither described, nor labeled. It would help the reader to provide more details.

l. 348: 'medial pterygoid-palatine suture...': In the description you mention that the palatines are not preserved. Then the suture with the pterygoid is probably difficult to describe (but see my comments below about the palate).

l. 351: 'fossae for both the preorbital and postorbital lobes': This is surprising for a platanistoid. This is not stated in the description, and in your discussion, you only suggest the presence of a fossa for the preorbital lobe, not for the postorbital lobe (l. 1011-1016). Diagnosis, description and discussion should be homogenized (ideally also with the codings in the phylogenetic analysis).

l. 355: 'lacking the medial surface of the falciform process...': Considering the preservation state of the specimen, it may be difficult to estimate if this suture was genuinely absent or if the condition results from postmortem break.

l. 358: 'weakly developed antorbital notch': The outline of the antorbital process is damaged on both sides. If you think that a reconstruction in dorsal or ventral view can be provided based on preserved elements, it would better support this character, otherwise not really convincing.

l. 379: 'a posterior dorsal infraorbital foramen...': No foramen is labeled near the posterior extremity of the premaxilla in fig. 2. Do you mean that this foramen is lacking here? I guess so. In that case the sentence should be made clearer in adding 'lacking' at the beginning. Considering the condition in other allodelphinids (including A. pratti, labeled in Kimura & Barnes, 2006, but not in your figure 7), I would tend to think that the lack of a posterior infraorbital foramen in Arktocara is related to preservation.
Note also that in the description (l. 522) you mention a posterior dorsal infraorbital foramen. Is it homologous to the posterior foramen close to the end of the premaxilla in many other odontocetes (and most allodelphinids)? Or do you think that it is a more anterior foramen?
I would suggest to always state if present/absent for each separate character in the diagnosis. In several cases the formulation is slightly confusing. Also, the use of ',' and ';' should be more consistent to avoid any confusion. See for example l. 381-382.


l. 540: 'which would have been overlaid by the missing palatines': I am not sure to understand how the palatine is missing if the pterygoid is at least partly preserved in the palate. Indeed, the latter partly covers the palatine in many, if not all, crown odontocetes. Therefore, at least a part of the palatine must be preserved under the pterygoid. In general, the pterygoid gets much more easily detached than the palatine. Would it be possible that the palatine is indeed preserved, except for the thin plate making the ventral wall of the palatine groove, and that the maxilla-palatine suture is not visible (which is often the case in fossil odontocetes)?

l. 542: 'and encircles both of the ventral infraorbital foramina': Do you mean that there are two ventral infraorbital foramina on each side? Could one of them be instead the sphenopalatine foramen?

l. 581: '1.9 cm in length': I would give this as a minimum estimate. The nasals could have slightly overhung the bony nares, and thus have been slightly longer (see for example the holotype of A. pratti).

l. 601-602: The posterior outline of the vomer in fig. 3 is surprising compared to many other odontocetes, including A. pratti. I would tend to think that the basisphenoid-basioccipital suture labeled in this figure is actually the posterior margin of the vomer. Just a suggestion.

l. 662-3: 'interrupted by a narrow cleft ... small knob...': This should be labeled on one of the figures. Could this small knob correspond to a vestigial medial crest of the exoccipital, as observed in basilosaurids, instead of a part of the basioccipital?

l. 701: 'that indicate where the palatines would have ...': I think that I see what you mean, but the palatal groove does not follow the lateral margin of the palatine. So the groove does not provide the original outline of the palatine, but rather a minimum extent of the palatine. + see my comment above about the loss of the palatine.

l. 705-706: 'The size of the palatine groove suggests that...': I do not follow you here. The palatine groove is a canal covered by the palatine, which gives an idea of the minimum extent of the palatine in the palate region. But it is not related to the presence of an extension of the palatine in the orbit region. These are two separate regions, and you can have an anteriorly long palatine without any significant posterior extension in the orbit region (e.g., beaked whales). Furthermore, there is some confusion here between the lateral lamina of the palatine and the lateral lamina of the pterygoid (see below). I would remove this section.

l. 728-730: 'lanceate process of the squamosal': How do you distinguish this process from the lateral lamina of the pterygoid? The suture between the falciform process and the pterygoid can be difficult to distinguish in fossils. And if this is indeed a part of the falciform process, does it really differ from the large process seen for example in Zarhachis? This is an interesting structure, but I am not sure if it needs a new name.

l. 732: 'lateral lamina of pterygoid': As mentioned above, I do not think that there is a link between the palatine groove and the posterior extent of the lateral lamina of the palatine or (in this paragraph) the pterygoid. I am much more convinced by your second argument (faciform process/lanceate process), most likely indicating a long lateral lamina of the pterygoid (not palatine).

l. 786: 'Lambert et al. (2014) recovered a paraphyletic Allodelphinidae': This is not the case in the 2014 paper (part of the tree unresolved, but see 50% majority consensus tree, with monophyly), but paraphyly is indeed obtained in Lambert et al. (2015. Chilcacetus).

l. 795: 'Our analysis is the first one to recover...': This is not true I think. Tanaka & Fordyce (2015a, fig. 21 top and 2015b, fig. 8a) did find this clade in part of their analyses.

l. 824: 'though the migration of the palatines ... can be inferred ...': In Arktocara, the palatine groove is far anterior to the presumed anterior end of the pterygoid. So the derived state is most likely not present based on the proposed interpretation (but see comments above on that interpretation).

l. 1126: 'for an exceptional occurrence of a platanistid in freshwater': There is also a record by Hulbert & Whitmore (2006) of one mandible of Pomatodelphis inaequalis from freshwater deposits of Alabama. This could be mentioned here.

Hulbert, R. C., and F. C. Whitmore, Jr. 2006. Late Miocene mammals from the Mauvilla Local Fauna, Alabama. Bulletin of the Florida Museum of Natural History 46(1):1-28.

·

Basic reporting

Language is generally ok. There are a few minor stylistic issues to fix, some indicated on the annotated pdf. Some cladistic characters could be identified more explicitly as present or lacking.

Introduction and background comments are helpful and relevant for phylogenetic-biosystematic vertebrate paleontology.

The literature is fine - a suitable mix of historically important and current.

The ms follows PeerJ structure.

Figures are highly relevant and the photos are technically of high quality in terms of lighting, crispness etc. However, I struggled to interpret some structures on the unlabelled and unmasked original photos because of the variable colours and irregularly patchy surfaces. In turn, the labelled masked figures have a murky tonal range which obscures details. This fossil would benefit hugely from being coated with sublimed ammonium chloride and photographed accordingly, as done recently for comparable fossils described and named in other publications (e.g. skulls of Papahu and Otekaikea). Yes, this is a fiddly technique, but with excellent results if done ok – each resulting picture is worth 1000 words. Note that I am not saying that the whitening method must be used, merely noting that standards have risen and readers might expect better. But, I think it would make the figs much more informative, leading to more reads and citations. (The labelling, also, is somewhat minimalist, and figs would be improved if more features were noted – but this is a personal preference.)

What page size and format will be used for final print? - I am curious because, at review size, the fine details are not clear, and the figs might thus be enlarged. Some specifics on labelling and presentation are detailed below. It is not clear why Figs 10a-d are included; they add little description to the ms, and they are too dark to be informative - even if they were labelled.

Raw data are supplied appropriately, in the form of figures, and phylogenetic scorings and character states.

Overall assessment: generally fine.

Experimental design

The ms is an original account that identifies, describes and interprets a hitherto unknown fossil dolphin which helps address hypotheses about the origins of the endangered living South Asian River Dolphin, Platanista. The authors describe their material in words rather than by extensive numerical data, using expected terminology to identify homologous structures; the written description is complemented by a cladistic character matrix which is used to produce a phylogeny that in turn allows wider interpretation.

The ms clearly identifies research questions about the identity and recognition of the clade Platanistoidea.

The research meets the expected standards of biosystematic study on a single yet highly informative specimen. Indeed, when revised, the text of this ms might usefully be followed by some others in the field. Methods are clearly outlined, and could easily be followed by other researchers. Nevertheless, there are issues to deal with through minor revision – see comments below and on the annotated pdf.

I feel that the descriptive text and accompanying figures are in combination weaker than the introductory sections and the discussions of systematics and cladistics of the Allodelphinidae. Yet, those latter sections are based on or relate to the former. I hope that the descriptions can be improved by clarifications and additions, and the figures improved by more labelling and edits of tonal range etc.

The study meets the prevailing ethical standards in the field of vertebrate paleontology.

Overall assessment: generally fine.

Validity of the findings

Data are generally robust and sound. I have some minor queries on homologies of some features, noted below and/or on the annotated review pdf.

All relevant data are provided.

In general, conclusions are clearly and appropriately stated, and shown to be relevant to the problems identified in the introductory sections. However, my attached comments suggest that there are features that should be mentioned (as present or absent) because of their distribution in other odontocetes – the present omissions stand out (eg structure of posterior apex of premaxillae, or premaxillary sulci). For some other structures, I query proposed homologies.

There is no unfounded speculation.

Decisions are made for scientific reasons.

The research could be replicated: phylogenetic aspects of the research could readily be tested by other researchers by adding other taxa and/or by revisiting the homology of some characters and/or by ordering phylogenetic characters, or using implied weighting.

Some results are inconclusive, mainly those indicated by polytomies on the phylogeny. Such patterns are extremely important in identifying problems needing more attention.

Overall assessment: generally fine.

Additional comments

See the annotated pdf for suggestions for alternative wordings. Note also these specific points:

L296 ... belongs to the node-based Platanistoidea based on one unequivocal synapomorphy: the alisphenoid-squamosal suture coursing along the groove for the mandibular branch of the trigeminal nerve in ventral view (character 147[1]).
This important, indeed key, character should be figured clearly; at present it is not. A photo would not be essential if high-quality line art were used.

L311 ... Arktocara differs from all platanistoids outside Allodelphinidae in having the ventral edge of the zygomatic process of the squamosal almost straight in lateral view (character 113[1]).
This seems at odds with the irregular profile as shown in Figs 4B and D. If the structure is indeed straight, explain why it looks irregular. Does it need to be photographed from another angle.

L340-357 This section details character states that Arktocara HAS, but some of the states appear to be ambiguously stated or maybe wrongly interpreted – at least as far as I can see from description and/or figures. Thus –

L344 the parietals in dorsal view completely fused to and indistinguishable from the frontals and supraoccipital …
Elsewhere the text is ambiguous about the absence, or presence and presumed fusion, of the parietal on the vertex. Resolve. Could help also to change the currently heavy line along the nuchal crest, which might be mistaken for a suture; try a dashed line. See also L969.

L349 medial pterygoid-palatine suture angled anteromedially in ventral view (character 126[0]);
The palatine is lost and the suture is not preserved. You can't say what state it is/was, or use it in diagnosis.

L351 ... fossae for both the preorbital and postorbital lobe of the pterygoid sinus present in the orbit (characters 136[1] and 136[1] respectively...
Duplicated character numbers. You don’t make a good case for pre- and postorbital sinus fossa in the written description, but I wonder about structures that I have circled and commented on in Fig 3B. See also comment under L1013, on equivocal preorbital fossa.

L357 … pneumatic maxillary crest overhanging medially (character 65[0]).
But Arktocara lacks a pneumatic crest; do you mean to say lacking?

L364. … posteroventral-most point of the basioccipital crest forming a closely appressed flange separated dorsally from the rest of the crest by a narrow crease.
In other early Neoceti, and in some living species (eg young Tursiops), a crease here marks the exoccipital-basioccipital suture; see Mead & Fordyce on basioccipital/pharyngeal crest. If this ID seems correct, suggest include it in description as well as in characters/diagnosis.

L376. Again – see L349 comment above, this char 126 is not preserved in Arktocara.

L398. … the transverse width of the nasal >70% the length of the nasal (character 91[2]);
How can you say this when the nasals are lost from Arktocara? If you can justify it in the descriptive section (namely, use fossa as a prediction for nasal size and proportions) then there may be better grounds.

L432. State if any burial-related distortion is apparent.

L504
Any hint of division of premaxilla into posteromedial splint and posterolateral plate? Given widespread occurrence of that feature amongst odonts, perhaps you should say yes/no.

Other important features to note as present (in that case give details) or absent: premaxillary foramen, and the normally-associated premaxillary sulci.

L522. … The posterior dorsal infraorbital foramina on both sides are v-shaped,…
The figs suggest that the foramina are single (?subcircular), and that it is the sulci arising from the foramina are V-shaped.

L533. … where it flattens out on the maxillary flange
Mead & Fordyce note that the maxillary flange is a dorsal feature associated with origin of rostral muscle. Is that what you mean?

L537. …the palatine groove of the maxilla begins approximately 1.5 cm lateral of the midline, and curves posterolaterally around the dorsal lamina of the pterygoid.
The photos suggest 2 features here: a narrow palatine groove or sulcus for vessels/nerves (sensu Mead & Fordyce) – often seen when the palatine is removed, and a larger bilateral fossa which very likely marks the maxillopalatine suture. Comparable structure occurs in Waipatia for example. You should revise to discriminate between a fossa for the suture, and the vascular/neural sulcus. See notes drawn on Fig3B.

L540. … Between the palatine groove and the medial lamina of the pterygoid is a fossa, which would have been overlaid by the missing palatines and housed the anterior pterygoid sinus…
Here is possible confusion. See above for interpretation of a fossa in this region as for maxillopalatine suture. More-posteriorly, flooring the nares, we would expect hamuli with hamular lobes of pterygoid sinus. For these features near nares, avoid the name anterior sinus, which is a projection from the preorbital sinus onto rostral base.

L573. …. The jugal is missing, but the jugular process of the lacrimal is preserved…
1) Presumably the lacrimal and jugal are fused - right? Or do you propose they are separate as in Ziphiidae? If the 2 elements are unfused then the jugal alone could be wholly lost.
2) You use the term jugular process – presumably to mean the base of the jugal. But, jugular process is a term for the paroccipital process.
3) It is unlikely that the base of the jugal is at the spot labelled jugular process on Fig 3. Rather, there a feature seen further anteromedially may be the base of the styliform process and thus strictly part of the jugal. See Fig 3 for notes

L662 … The posterior end of the basioccipital crest is interrupted by a narrow cleft that separates a small knob from the rest of the tympanic plate….
See L364, above. In many cetacea including fossil and modern young animals, the exoccipital contributes to posterior of pharyngeal crest. See Mead Fordyce for more.

L671 Sphenoid
Comment on a so-far unmentioned component of sphenoid cluster, namely orbitosphenoid. Hard to tell but in fig 3 there seems to be structure in anterior wall of orbit. The ethmoid foramen is a giveaway. Whether identifiable or not, you should say.


L683
In many Cetacea, including basilosaurids and neocetes, a wedge of alisphenoid extends backwards to bound the medial edge of the falciform process. It would help to see if present here.
See also L724

L701
See comments for L537

L725 ff. Text is difficult to match with figures. Suggest revisit it in light of pterygoid sinus fossa structure – see comments below for L732.

L732 ff. I feel you might discuss the pterygoid sinus fossa/e better, given the importance of the pterygoid sinus in biosystematics. Was there likely a complete lateral lamina of pterygoid (possibly with palatine contribution) as I’ve queried on Fig 3B, or have I mistaken the feature circled (in which case, others might too)? The latter structure looks plausble for elongate, bony wall of a sinus fossa. What about the possible post and preorbital fossae? – what are the features labelled on Fig 3B? - does the skull show whether there was a roof to the pter sinus fossa, and/or is there evidence that the sinus perforated that roof to rise dorsally into the orbit sensu lato? These points would be addressed well by providing a labelled photo of the skull coated with sublimed ammonium chloride.

L799
The discussion from here on is clear and informative, with only minor points to query. Possibly some parts will need rewrite in light of questions of homology raised above.

L824. … though the migration of the palatines dorsolaterally can be inferred directly from the palatine groove of the maxilla
Note earlier comments on palatine groove and possible structure of the palatine. Specimens of basal platanistoids as recently recognised – species of Waipatia and Otekaikea – don’t provide much evidence about palatine structure.

L918. … alisphenoidal-squamosal suture coursing along groove for mandibular branch of trigeminal nerve in ventral view (character 147[1]
Given the importance of this feature, mark it if possible on Fig 6A or B.

L1013 ...However, in Arktocara yakataga, though there is no obvious indication of a postorbital lobe of the pterygoid sinus, the deep and broad fossa surrounding the ventral infraorbital foramina suggests the presence of a preorbital lobe...
See comments above at L351, 732.

Figures
Some of these have queries about interpretation
Figure 10 adds little of value, beyond showing that the specimens exist.

Fig 11, cladogram
This begs the question: why propose a new genus for Arktocara yakataga? Why not Allodelphis yakataga?

Supplement 2 on phylogenetics
I have browsed this but have not the time to review it closely.

Reviewer 3 ·

Basic reporting

I have few comments here. My only confusion relates to the age of the specimen. I understand that the exact locality is unknown, and that the Poule Creek Formation spans a lot of time, but I had trouble getting a sense of what the possible age range of the specimen is, and how its stratigraphic position relates to dates from detrital zircons. Although unlikely, could the specimen by early Miocene if sedimentation rates were really variable? The uncertainty/range needs to be more clearly explained.

Minor comments are included on the annotated pdf.

Experimental design

My primary problem with this manuscript is the absence of molecular data from the phylogenetic analysis. If a clade or grade consists of only extinct taxa (for example determining the phylogeny of sauropod dinosaurs or protocetid whales) then it is probably fine to not include molecular data for the closest extant taxa. However, this is not the case here. Some of your primary findings relate to the age of the odontocete crown group; however, our current understanding of the topology of extant taxa is really based on molecular data. For example, molecular data supports physeteroids as the basal extant odontocete clade, followed by Platanista, next Ziphiidae, and then Delphinida. This is quite different than the relationships you obtained, and I am very concerned that your results would change if molecular data are included. There really is no justification for excluding molecular data.

I can see three ways you could address this problem. 1) Scale back the findings of your paper to just focus on the new taxon and its position within Allodelphinidae. That result is well supported by your analysis and I doubt will change in future analysis. 2) You could include molecular data in your analysis. This would be best but is a big job, so a more efficient option would be 3) to do an analysis with a molecular backbone constraint to mimic the effect of including molecular data

Validity of the findings

The findings are generally valid, except those that relate to age of the crown group (see comments in section above).

I would also strongly discourage you from naming Pan-Platanista, or if you choose to retain it, keeping/redefining Platanistoidea. The distinction between the two taxa is quite subtle, and in fact, most authors use Platanistoidea in the same way as you define Pan-Platanista. I am also concerned about the inclusion of Waipatia in the definition for Platanistoidea. What if Waipatia falls outside the crown group. Do we them abandon it, or does it then refer to a much more inclusive clade then originally intended? In short, the relationships of Platanista and extinct relatives are quite unstable, so this may not be the best time to name new clades in that part of the tree.

Additional comments

This is an excellent paper and one that I hope to see published. Your identification and description of the specimen look solid, and I have no doubt that it is a new taxon and that it is a allodelphinid. I do feel strongly about the incorporation of the molecular data (or equivalent signal) in your phylogenetic analysis. A backbone constraint is not that hard to use, and I would encourage you to use that option. It will likely yield a result that is not only more accurate, but will likely be more consistent with future total evidence analyses.

Annotated reviews are not available for download in order to protect the identity of reviewers who chose to remain anonymous.

---

## Round 0.2 · Minor Revisions

This manuscript is in excellent shape and the authors have addressed the comments from reviewers and myself diligently. It is nearly ready to be accepted. There are just a couple of near cosmetic issues to be taken care of. These do not require additional review.
1. I found two 'Platanista' that are not italicized. Search your ms for 'fossil relatives of Platanista" and for 'most closely related to Platanista' to find them.
2. Beitrage zur Palaontologie, You dropped the umlauts in this reference. I thought there was one on both Beitrage and Palaontologie. Please check.
3. Museum National d'Histoire Naturelle, the first word here does have -um and an accent on the e in the d'Orbigny reference. The Muizon 1991 reference has it spelled correctly but also misses the accent
4. Two of the Muizon references do not have the journals in italics.
3. The Joyce reference has paleontology misspelled.
Otherwise, great job.

·

Basic reporting

ok

Experimental design

ok

Validity of the findings

ok

Additional comments

The authors made the effort to modify parts of the diagnosis, description, ilustrations, and even codings for the phylogeny as suggested by the reviewers, and they could answer satisfactorily to all the queries I noted. This paper certainly deserves to be published as it is in PeerJ. As a final (and very minor) comment, note that if Otekaikea was not included in Waipatiidae in the trees by Tanaka & Fordyce (2015a,b), the species Otekaikea marplesi falls indeed in Waipatiiae in figure 19 of Tanaka & Fordyce (2014).

O. Lambert

---

## Round 0.3 · accepted · Accept

Thanks for your submission, this is a significant contribution to cetacean evolution. I look forward to speaking with you about this sometime.